# Latent Zoning Network:
# A Unified Principle for Generative Modeling, Representation Learning, and Classification

**Zinan Lin**[*]
Microsoft Research
Redmond, WA, USA

**Enshu Liu**
Tsinghua University
Beijing, China

**Xuefei Ning**
Tsinghua University
Beijing, China

**Junyi Zhu**
KU Leuven
Leuven, Belgium

**Wenyu Wang**
Redmond, WA, USA

**Sergey Yekhanin**
Microsoft Research
Redmond, WA, USA

## Abstract

Generative modeling, representation learning, and classification are three core problems in machine learning (ML), yet their state-of-the-art (SoTA) solutions remain largely disjoint. In this paper, we ask: *Can a unified principle address all three?* Such unification could simplify ML pipelines and foster greater synergy across tasks. We introduce `Latent Zoning Network` (`LZN`) as a step toward this goal. At its core, `LZN` creates a shared Gaussian latent space that encodes information across all tasks. Each data type (e.g., images, text, labels) is equipped with an encoder that maps samples to *disjoint latent zones*, and a decoder that maps latents back to data. ML tasks are expressed as compositions of these encoders and decoders: for example, label-conditional image generation uses a label encoder and image decoder; image embedding uses an image encoder; classification uses an image encoder and label decoder. We demonstrate the promise of `LZN` in three increasingly complex scenarios: **(1)** `LZN` **can enhance existing models (image generation):** When combined with the SoTA Rectified Flow model, `LZN` improves FID on `CIFAR10` from 2.76 to 2.59—without modifying the training objective. **(2)** `LZN` **can solve tasks independently (representation learning):** `LZN` can implement unsupervised representation learning without auxiliary loss functions, outperforming the seminal MoCo and SimCLR methods by 9.3% and 0.2%, respectively, on downstream linear classification on `ImageNet`. **(3)** `LZN` **can solve multiple tasks simultaneously (joint generation and classification):** With image and label encoders/decoders, `LZN` performs both tasks jointly by design, improving FID and achieving SoTA classification accuracy on `CIFAR10`. The code and trained models are available at https://github.com/microsoft/latent-zoning-networks. The project website is at https://zinanlin.me/blogs/latent_zoning_networks.html.

## 1 Introduction

Generative modeling, representation learning, and classification are three of the most widely used machine learning (ML) tasks. Generative models like DALLE [71, 70, 5] and GPT [68, 69, 7, 1] power applications such as question answering and content creation. Representation learning, exemplified by CLIP [67], supports tasks like information retrieval. Classification is central to tasks such as object recognition [17] and sentiment analysis [19, 54].

Notably, the state-of-the-art (SoTA) techniques for these tasks differ. For example, SoTA generative modeling relies on diffusion models [32, 76, 78] and auto-regressive transformers [68, 69, 7, 1]; SoTA

---

[*]Correspondence to: Zinan Lin (zinanlin@microsoft.com).

39th Conference on Neural Information Processing Systems (NeurIPS 2025).

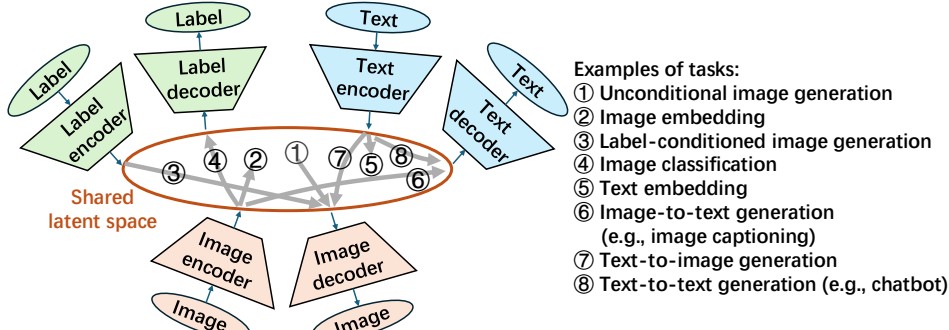

Examples of tasks:
① Unconditional image generation
② Image embedding
③ Label-conditioned image generation
④ Image classification
⑤ Text embedding
⑥ Image-to-text generation
  (e.g., image captioning)
⑦ Text-to-image generation
⑧ Text-to-text generation (e.g., chatbot)

Figure 1: `Latent Zoning Network` (`LZN`) connects multiple encoders and decoders through a shared latent space, enabling a wide range of ML tasks via different encoder-decoder combinations or standalone encoders/decoders. The figure illustrates eight example tasks, but more could be supported. Only tasks 1-4 are evaluated in this paper, while the rest are for illustration.

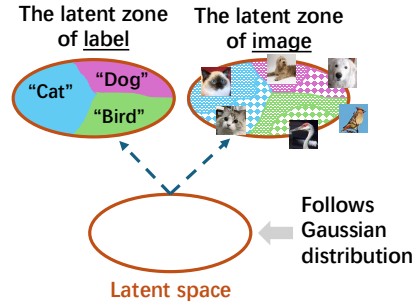

Figure 2: The latent space of `LZN` has two key properties: **(1) Generative:** It follows a simple Gaussian prior, allowing easy sampling for generation tasks. **(2) Unified:** It serves as a shared representation across all *data types* (e.g., image, text, label). Each data type induces a distinct partitioning of the latent space into *latent zones*, where each zone corresponds to a specific sample (e.g., an individual image or label). The latent space is shown as a closed circle for illustration, but it is unbounded in practice.

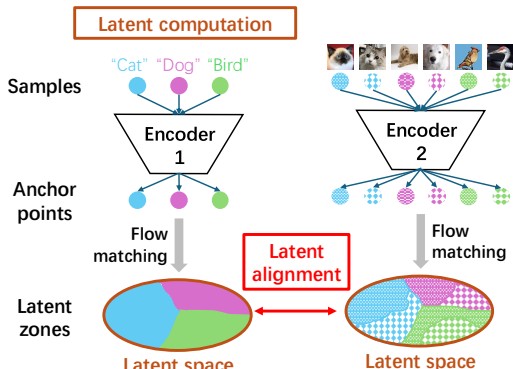

Figure 3: Training and inference in `LZN` rely on two atomic operations: **(1) Latent computation (§ 2.2.1):** Computes latent zones for a data type by encoding samples into *anchor points* and using flow matching (FM) [53, 51] to partition the latent space. Conversely, any latent point can be mapped to a sample via the decoder (not shown). **(2) Latent alignment (§ 2.2.2):** Aligns latent zones across data types by matching their FM processes. *This figure also illustrates the approach for `LZN` in joint conditional generative modeling and classification (§ 5).*

representation learning employs contrastive loss [28, 10, 26]; and SoTA classification uses dedicated models trained with cross-entropy loss and its variants [46]. Although using distinct methods for these tasks has long been established and widely accepted in the community, we revisit this methodology from first principles and question its necessity. Specifically, we ask, out of curiosity:

*Can a single principle unify generative modeling, representation learning, and classification?*

Part of our motivation stems from Occam's Razor [83], which favors simpler solutions when possible. More importantly, while these tasks differ in formulation, they are fundamentally related and can benefit from one another; a unified principle could facilitate such synergy.[2]

In this paper, we reflect on the strengths and limitations of existing techniques and propose a new unified framework, `Latent Zoning Network` (`LZN`), illustrated in Figs. 1 and 2. At the core of our design is a shared *latent space* that connects a series of encoders and decoders, each corresponding to a specific *data type* (e.g., images, text, labels). Encoders map data into a *zone* in the latent space, and the corresponding decoder maps it back to the data. Different tasks can be interpreted as

---

[2]While auto-regressive (AR) transformers with large-scale pre-training provide one approach to unify these tasks [68, 69, 7, 1], SoTA transformer-based representation learning still relies on contrastive learning [45, 89]. More importantly, our approach can be viewed as an orthogonal layer on top of transformers and should be seen as complementary rather than competing. See § 2.4 for further discussion.

performing "translations" within the latent space—either using encoder–decoder pairs or leveraging a single encoder or decoder. Compared to popular *representation learning* approaches, which place no constraint on the latent distribution [10], LZN's latent space is *generative*: it follows a simple prior distribution for easy sampling. In contrast to modern *generative modeling* approaches, where different conditions (e.g., class labels, text) are treated as separate inputs [87], LZN maintains a *single* latent space that unifies different types of conditional information. Finally, unlike standard *classification*, where class labels are model outputs, LZN treats "class labels" as a data type connected through the latent space like any other data type.

To train and perform inference with LZN, we rely on two atomic operations (Fig. 3 and § 2): **(1) Latent computation:** Given an encoder, we compute the *latent zones* for a batch of samples. Specifically, we first use the encoder to compute each sample's *anchor point*, then apply flow matching [53, 51] to map these points to their corresponding latent *zones*. This procedure ensures that the resulting zones collectively follow a simple Gaussian distribution, facilitating generation tasks, while also guaranteeing that zones from different samples remain disjoint–allowing them to serve as unique latents for classification and representation learning. **(2) Latent alignment:** Aligning the *latent zones* produced by two different encoders to facilitate the tasks that require translations between encoders and decoders from different data types. This is a fundamentally challenging task due to its discrete nature. To address it, we introduce a novel "soft" approximation that performs the alignment midway through the flow matching process, enabling effective and tractable training.

We demonstrate that, despite its simplicity and reliance on just two atomic operations, LZN is capable of supporting a wide range of seemingly diverse tasks. To illustrate its versatility and practical utility, we present three *levels* of applications:

- **L1: Enhancing *one* existing task (§ 3).** Because LZN latents can be computed without supervision, they can be seamlessly integrated into existing models as an additional conditioning signal–without requiring any changes to the training loss or methods. In this setup, LZN latents hopefully can learn to improve the task performance. To demonstrate this, we incorporate LZN into rectified flow models [53]–a state-of-the-art generative approach for images–and observe improved sample quality across CIFAR10, AFHQ-Cat, CelebA-HQ, LSUN-Bedroom datasets. Specifically, on CIFAR10, LZN closes the FID gap between conditional and unconditional generation by 59%.
- **L2: Solving *one* task *independently* (§ 4).** LZN can also tackle tasks entirely on its own, without relying on existing methods. As a case study, we use LZN to implement unsupervised representation learning, a task traditionally addressed with contrastive loss. We find that LZN can even outperform the seminal methods such as MoCo [28] and SimCLR [10] by 9.3% and 0.2%, respectively, on downstream ImageNet linear classification.
- **L3: Solving *multiple* tasks *simultaneously* (§ 5).** Pushing further, LZN is capable of handling *multiple* tasks at once. In particular, we employ two encoder–decoder pairs—one for images and one for labels—enabling LZN to jointly support class-conditional generation and classification within a single, unified framework. Built on rectified flow, this implementation outperforms the baseline conditional rectified flow model in generation quality, while also achieving state-of-the-art classification accuracy. Notably, the performance on both generation and classification exceeds that of training each task in isolation. This supports our core motivating intuition: seemingly distinct tasks can benefit from shared representations, and LZN provides a principled framework for enabling such synergy.

While the early results are promising, many challenges, open questions, and exciting opportunities remain unexplored. In principle, as more encoder–decoder pairs are added to the shared latent space, the range of applications LZN can support should grow at least quadratically (Fig. 1). Whether LZN can scale gracefully and realize this potential remains to be seen. We hope this work opens a new line of research toward this ambitious vision. See more discussions in § 6.

## 2 Latent Zoning Network (LZN)

### 2.1 Overall Framework

**Revisiting existing approaches.** To motivate our design on a unified framework for diverse ML tasks, we first analyze the strengths and limitations of existing approaches.

- **Generative modeling.** Given samples $x \sim p$ from an unknown distribution $p$, generative models aim to learn a decoder $D_x$ such that $D_x(z)$ approximates $p$, where $z$ is random noise drawn

from a simple distribution.[3] While $z$ can carry useful information for representation learning and classification [50], this pipeline has key limitations: **(1)** The mapping from $z$ to $x$ lacks flexibility. For example, in diffusion models, the optimal mapping is fixed once the distributions of $x$ and $z$ are fixed [78]. To introduce controllability, models often augment $D$ with additional condition inputs $c_1, \ldots, c_k$: $G(z, c_1, \ldots, c_k)$ [87]. This is suboptimal–conditions may overlap or conflict (e.g., text vs. label conditions in image generation), and the resulting representation $(z, c_1, \ldots, c_k)$ becomes fragmented. **(2)** Inverting $D$ to recover $z$ from a sample $x$ is non-trivial for some SoTA generative models [36]. These issues limit the effectiveness of generative models for representation learning, as also observed in prior work [24].

- **Unsupervised representation learning.**[4] SoTA representation learning typically uses contrastive loss [10], where an encoder $E$ maps related sample pairs–either from the same modality (e.g., image augmentations) or across modalities (e.g., image–text pairs)–to similar embeddings, while pushing unrelated samples apart. These embeddings can perform zero-shot classification by comparing pairwise cosine similarities [67] or be adapted for classification using a linear head trained with cross-entropy loss [10]. However, contrastive loss leads $E$ to discard important details (e.g., augmentations, modalities), making the representations unsuitable for standalone generation. Moreover, with few exceptions [2], the representations lack distributional constraints, making them hard to sample from for generative tasks unless training an additional generative model [44].

- **Classification.** The most common and SoTA classification approach trains a dedicated model with cross-entropy loss to map inputs to class labels [19, 54]. Intermediate layer outputs can be used as representations [80]. However, because the objective focuses solely on classification, these representations tend to discard class-irrelevant information, limiting their utility for general-purpose representation or generation. As with contrastive learning, they also lack distributional constraints for generative tasks.

While one could combine the above objectives and methods into a single training setup [60, 43], our focus is on designing a clean, unified framework that naturally integrates all these tasks.

**Desiderata.** We observe that all the above tasks can be framed as *learning mappings between data and a latent space*. The main differences lie in: the mapping direction (e.g., latent-to-data for generation, data-to-latent for representation/classification), constraints on the latent space (e.g., a simple prior for generative models, none for others), and the amount of information encoded (e.g., class labels for classification tasks, detailed reconstructions for generative models). To support all these tasks in a single framework, we seek: **(1) A unified latent space** that captures all necessary information of all tasks; **(2) A generative latent space** that follows a simple distribution; and **(3) Easy mappings** between data and latent in both directions.

**Framework.** To address the above desiderata, our key designs are (Fig. 2):

- **A unified latent space.** In existing frameworks, a sample like text can play inconsistent roles–appearing as input latent in text-to-image generation or as output in text generation. This makes it hard to define a unified latent space across tasks.

  We address this by introducing a hypothetical *foundation latent space* that represents all possible samples in the world. Each foundation latent is an abstract entity that appears through *observations* in different *data types*, such as images, text, and even class labels (e.g., "cat" or "dog").

  Importantly, different latents can share the same observation (e.g., multiple cat images all labeled "cat" and described as "a cat image"). As a result, each observed sample defines a *latent zone—a* subset of the latent space that produces the same observation in that data type. This provides a unified way to represent and connect all data types within the same latent space.

- **A generative latent space.** We enforce the latent space to follow a Gaussian distribution, enabling easy *unconditional* sampling without constraining any data type. Our framework also supports easy *conditional* sampling from a *latent zone* induced by an observed sample (e.g., a label).

- **Easy mappings.** Given samples of a data type, we compute their latent zones via the corresponding *encoder*. Conversely, a latent point can be decoded into a data type using its *decoder*.

**Tasks.** This design naturally supports a variety of tasks (Fig. 1):

---

[3]For diffusion models [32, 76, 78], $z$ is the initial Gaussian noise in the sampling process, plus intermediate noise if using SDE sampling [79]. For AR transformers, $z$ can be seen as the randomness in token sampling.

[4]We use "latent", "representation", and "embedding" interchangeably in the paper.

- **Single-module tasks.** A standalone encoder or decoder can perform specific tasks independently. For instance, the image encoder alone produces image embeddings (representations), while the image decoder alone enables unconditional image generation.
- **Cross-module tasks.** Any encoder–decoder pair defines a task. For example, `label encoder + image decoder` enables class-conditional image generation, `image encoder + label decoder` does classification, and `text encoder + text decoder` supports text generation.

We expect that tasks can benefit from each other through this unified framework (validated in § 5). Each task contributes its core information to the latent space, making it increasingly expressive and powerful. Conversely, since all tasks interface through the same latent space, improvements in the latent representations can facilitate learning across tasks.

As the latent space partitions into zones, we name this framework `Latent Zoning Network` (`LZN`).

## 2.2 Implementation of Atomic Operations

Training and inference in `LZN` rely on two operations (Fig. 3): *latent computation* and *latent alignment*. We will see in § 3 to 5 that these two operations are sufficient to implement a variety of tasks.

### 2.2.1 Latent Computation

**Desiderata.** Latent computation is important in both training and inference of `LZN`. Given samples $\mathcal{X} = \{x_1, \ldots, x_n\}$ of the same data type (e.g., images, text, labels), the goal is to sample their latents $z_1, \ldots, z_n = C(\mathcal{X})$ with a *random* latent computation function $C$ such that: **(1) Prior distribution is Gaussian:** the latent $z \sim \text{Uniform}\{z_1, \ldots, z_n\}$ follows Gaussian distribution $\mathcal{N}(0, \mathbf{I})$, and **(2) The latent zones of different samples are disjoint:** $\text{Supp}(z_i) \cap \text{Supp}(z_j) = \emptyset$ for $i \neq j$.

**Approach.** To achieve this, we first use a deterministic encoder $E_x$ to map each sample to an *anchor point* $a_i = E_x(x_i)$. We then apply the seminal flow matching (FM) method [53, 51], which establishes a one-to-one mapping between distributions, to transform these anchor points into latent zones. Specifically, we define a family of distributions $\pi_t$ with endpoints $\pi_0 = \mathcal{N}(0, \mathbf{I})$, the desired prior latent distribution, and $\pi_1(s) = \frac{1}{n} \sum_{i=1}^{n} \delta(s - a_i)$, the distribution of the anchor points.[5] The intermediate distribution $\pi_t$ is induced by linearly interpolating between samples $s_0 \sim \pi_0$ and $s_1 \sim \pi_1$ via $\varphi(s_0, s_1, t) = (1 - t)s_0 + ts_1$ (i.e., $\pi_t$ is $(1 - t)\pi_0 + t\pi_1$). The velocity field in FM [53] can then be computed as

$$V(s, t) \triangleq \mathbb{E}_{s_0 \sim \pi_0, s_1 \sim \pi_1} \left( \frac{\partial \varphi(s_0, s_1, t)}{\partial t} \middle| \varphi(s_0, s_1, t) = s \right) = \frac{\sum_{i=1}^{n}(a_i - s) \exp\left(-\frac{(s - ta_i)^2}{2(1-t)^2}\right)}{(1 - t)\sum_{i=1}^{n} \exp\left(-\frac{(s - ta_i)^2}{2(1-t)^2}\right)}.$$

We can obtain $s_t$ by integrating along the FM trajectory: $s_t = \text{FM}_x(s_0; t) \triangleq s_0 + \int_{\tau=0}^{t} V(s_\tau, \tau)\mathrm{d}\tau$ for $s_0 \sim \pi_0$. It has been shown [53] that the distribution of $s_t$ is $\pi_t$. Similarly, integrating backward $s_t = \text{IFM}_x(s_{1-g}; t) \triangleq s_{1-g} + \int_{\tau=1-g}^{t} V(s_\tau, \tau)\mathrm{d}\tau$ for $s_{1-g} \sim \pi_{1-g} \triangleq (1 - g)\pi_1 + g\mathcal{N}(0, \mathbf{I})$ also yields $\pi_t$, where $g$ is a small constant.[6] With a slight abuse of notation, we also write $s_t = \text{IFM}_x(a_i, \epsilon_i; t)$ to represent $\text{IFM}_x(s_{1-g}; t)$ with $s_{1-g} = (1 - g)a_i + g\epsilon_i$. With these setups, we define the latent computation as

$$z_i = C(\mathcal{X})_i \triangleq \text{IFM}_x(a_i, \epsilon_i; 0), \quad \text{where } \epsilon_i \sim \mathcal{N}(0, \mathbf{I}). \tag{1}$$

Due to the discussed FM properties, we can see that this approach (approximately) satisfies the two desiderata by construction (see App. A.1 for more details).

**Implementation.** Computing $C$ involves an integral, which we approximate using standard numerical solvers such as Euler or DPM-Solver [55, 56] that operates on a finite list of time steps $t_1, \ldots, t_r$, as in prior work on diffusion and RF models [53, 79]. A key property of our approach is that all operations are differentiable, allowing gradients to backpropagate all the way from latent $z_i$ to the encoders $E_x$ during training. More details are deferred to App. A.2.

**Efficiency optimization.** Computing the velocity $V$ requires access to all samples, making the memory and computation cost of $C$ high. To address this, we introduce several optimization

---
[5]$\delta$ denotes the Dirac delta function.

[6]FM is well-defined only when both $\pi_0$ and $\pi_1$ have full support. However, in our case, $\pi_1$ is a mixture of Dirac deltas and lacks full support. Therefore, we use the full-support distribution $\pi_{1-g}$ as the starting point.

techniques–latent parallelism, custom gradient checkpointing, and minibatch approximation–that make the training of LZN scalable. See App. A.3 for details.

### 2.2.2 Latent Alignment

**Desiderata.** Following § 2.2.1, latent zones from different data types are computed independently, which undermines the purpose of a shared latent space. Many applications require these zones to be *aligned*. We consider two types of alignment: **(1) Many-to-one (and one-to-many) alignment:** for example, the latent zone of the "cat" label should cover all latent zones of all cat images. **(2) One-to-one alignment:** for example, in image-text datasets, paired image and text samples should share the same latent zone. Concrete examples will be shown in § 4 and 5.

Formally, let $\mathcal{X} = \{x_1, \ldots, x_n\}$ and $\mathcal{Y} = \{y_1, \ldots, y_m\}$ be two datasets from different data types. The pairing is defined by $k_i$, where $y_i$ (e.g., a cat image) is paired with $x_{k_i}$ (e.g., the "cat" label). We aim to ensure $\text{Supp}\left(C\left(\mathcal{X}\right)_{k_i}\right) \supseteq \text{Supp}\left(C\left(\mathcal{Y}\right)_i\right)$ for all $i \in [m]$, meaning the latent zone of $x_{k_i}$ covers that of $y_i$. This formulation supports many-to-one alignments directly. For one-to-one alignment, a symmetric constraint can be added with $x$ and $y$ swapped.

**Approach.** Given the FM integral trajectories, alignment reduces to ensuring that the latent of $y_i$, when mapped via the trajectory, matches the anchor point of $x_{k_i}$: $\text{FM}_x\left(C\left(\mathcal{Y}\right)_i; 1\right) = E_x\left(x_{k_i}\right)$.

Challenge: discrete assignment is non-differentiable. Before introducing our solution, we illustrate why the problem is nontrivial by examining strawman approaches. A natural idea is to directly minimize the distance: $d\left(\text{FM}_x\left(C\left(\mathcal{Y}\right)_i; 1\right), E_x\left(x_{k_i}\right)\right)$, where $d\left(\cdot, \cdot\right)$ is a distance metric. This approach fails because FM deterministically maps each latent to exactly one anchor point $E_x\left(x_j\right)$, so the above objective effectively becomes minimizing the distance between anchor points. However, a latent zone is influenced by all anchor points, not just its own. Therefore, reducing the distance between a pair of anchors does not necessarily improve zone-level alignment. More fundamentally, the core challenge is that FM induces a *discrete assignment*: each latent deterministically maps to one anchor. This discrete operation is non-differentiable and cannot be directly optimized during training.

Technique 1: Soft approximation of alignment. To address this issue, our key idea is to introduce a soft approximation of the discrete anchor assignment process. Let us define $s_t^i = \text{FM}_x\left(C\left(\mathcal{Y}\right)_i; t\right)$ and $a_l = E_x\left(x_l\right)$. By construction, the distribution $\pi_t$ is a mixture of Gaussians: $\pi_t = \frac{1}{n}\sum_{l=1}^n \mathcal{N}(ta_l, (1-t)^2 \mathbf{I})$, where the $l$-th component corresponds to anchor $a_l$. We define the (soft) probability that $s_t^i$ is assigned to $a_l$ as being proportional to the density of $s_t^i$ under the $l$-th Gaussian component:

$$\mathbb{P}\left(a_l|s_t^i\right) = \exp\left(-\frac{\|s_t^i - ta_l\|^2}{2(1-t)^2}\right) / \sum_{j=1}^n \exp\left(-\frac{\|s_t^i - ta_j\|^2}{2(1-t)^2}\right).$$

This formulation provides a smooth, differentiable approximation of the otherwise discrete assignment. When $t = 0$, the approximation is fully smooth, with $\mathbb{P}\left(a_l|s_0^i\right) = 1/n$ for all $i, l$, reflecting a uniform assignment. As $t$ increases toward 1, the assignment becomes sharper. In the limit as $t \to 1$, it converges to the true discrete assignment, where $s_t^i$ deterministically maps to its assigned anchor.

From this, a straightforward idea is to maximize the assignment probability over all time steps such as $\sum_{t \in \{t_1, \ldots, t_r\}} \mathbb{P}\left(a_{k_i}|s_t^i\right)$ (recall that $t_i$s are solver time steps; see § 2.2.1). However, our ultimate goal is only to ensure correct assignment at $t = 1$. Even if this is achieved, the above objective would continue to push the intermediate states $s_t$ toward $a_{k_i}$, which is unnecessary and potentially harmful.

Technique 2: Optimizing maximum assignment probability. To avoid this, we propose to maximize $\max_{t \in \{t_1, \ldots, t_r\}} \mathbb{P}\left(a_{k_i}|s_t^i\right)$. This ensures that once the trajectory reaches the correct anchor near $t = 1$, the objective is maximized (i.e., equals 1) and no further gradient is applied as desired.

Technique 3: Early step cutoff. However, this approach introduces a new issue: if $s_t$ diverges from $a_{k_i}$ early on, the maximum probability remains at the constant $1/n$ (attained at $t = t_1 = 0$), yielding no training signal. To mitigate this, we truncate the set of time steps used in the maximization, restricting it to the later stages of the trajectory: $\{t_u, \ldots, t_r\}$, where $u$ is a hyperparameter that excludes early time steps. Putting it all together, our proposed alignment objective is:

$$\text{Align}\left(\mathcal{X}, \mathcal{Y}\right) \triangleq \text{maximize} \sum_{i=1}^m \max_{t \in \{t_u, \ldots, t_r\}} \mathbb{P}\left(a_{k_i}|s_t^i\right). \tag{2}$$

Please see App. B for more implementation details.

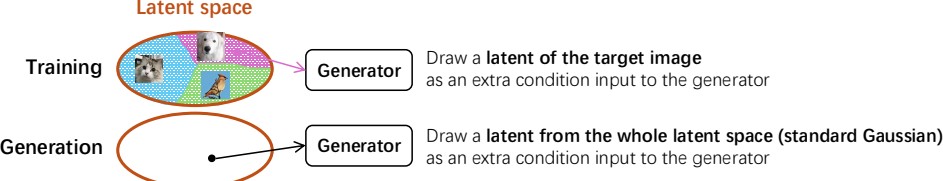

Figure 4: LZN for unconditional generative modeling (§ 3). During training, the LZN latent of each target image is fed as an extra condition to the rectified flow (RF) model [53], making the RF learn *conditional* flows based on LZN latents. The objective remains the standard RF loss, and the LZN encoder is trained end-to-end within it. During generation, we sample LZN latents from a standard Gaussian and use them as the extra condition. We illustrate the approach with RF, but since LZN latents require no supervision and are differentiable, the method could apply to other tasks by adding a condition input for LZN latents to the task network.

## 2.3 Decoder

The decoder $D_x$ maps the LZN latent back to its corresponding sample: $D_x(z_i) = x_i$. As we will describe in more detail later, it can be implemented using either a generative model (§ 3) or FM (§ 5).

## 2.4 Relationships to Alternatives

A prominent alternative that can also unifies different ML tasks is the use of AR transformers with large-scale pertaining, such as large language models (LLMs) [68, 69, 7, 1]. These models unify *generation* tasks by representing all data as sequences of tokens and modeling them in an AR manner. *Classification* tasks are cast as generation problems—for instance, prompting the model to complete the sentence "Which animal is in this image?" [69]. Additionally, prior work has shown that the intermediate outputs in these models can serve as strong *representation* for downstream tasks [72].

As such, this approach is *generation-centric*: the core formulation remains a generative modeling problem. Tasks that can be framed as generation–such as classification-can be unified naturally. However, for other tasks like representation learning, this approach must rely on surrogate methods, such as using intermediate model outputs. In contrast, *LZN offers a new formulation that seamlessly unifies all these tasks within a single framework*, as we will demonstrate in the following sections.

**More importantly, AR transformers (and other generative models) should be seen as *orthogonal and complementary* to LZN, rather than as competitors.** In particular, LZN decoders that map latents to data samples can be instantiated using any generative model. We demonstrate this in § 3 and 5. This allows LZN to leverage the strengths of existing generative models within its unified framework.

## 2.5 Scope of Experiments

While the framework is general, this paper focuses specifically on the *image* domain. We present three case studies: (1) generation (§ 3), (2) unsupervised representation learning (§ 4), and (3) joint classification and generation (§ 5). These case studies are arranged in order of increasing complexity: the first enhances a single task, the second solves a task using only LZN *without* any other external objectives, and the third tackles *multiple* tasks simultaneously within the same framework.

## 3 Case Study 1: Unconditional Generative Modeling

**Approach (Fig. 4).** Since LZN latents $C(\mathcal{X})$ can be computed using a standalone encoder without introducing new training objectives (§ 2.2.1), they can be easily integrated into existing pipelines as additional network inputs, without modifying the original loss function. We apply this idea to *unconditional generative models*, where the generator serves as the LZN decoder. We modify the decoder by adding an extra input to accept the LZN latents. Both the encoder and decoder are trained jointly using the original generative loss. In this setup, latent alignment (§ 2.2.2) is not used. Importantly, the encoder is only used during training to compute $C(\mathcal{X})$. During inference, latents are sampled from the Gaussian prior and passed directly to the decoder, **maintaining the original model's inference efficiency.** Please see App. C.1 for detailed pseudocode of the training and generation processes.

**Why it helps.** LZN latents provide unique representations for each image. This pushes the generator (decoder) to depend more directly on the latent for reconstruction, making the image distribution conditioned on the latent more deterministic. As a result, the generative objective becomes easier to optimize. Our experiments later confirm that LZN latents indeed capture useful features for generation.

Table 1: Unconditional image generation quality scores across four datasets. The best results are in `gray box`. Applying LZN to generative models improves RF on most image quality metrics. RF is a SoTA method; due to space constraints, we omit additional methods—see [53] for extensive comparisons between RF and others. Note that Inception Score (IS) is best suited for natural images like `CIFAR10`, though we report it for all datasets for completeness.

| Algo. | CIFAR10 ($32 \times 32$) | | | | | | | AFHQ-Cat ($256 \times 256$) | | | | | | |
|---|---|---|---|---|---|---|---|---|---|---|---|---|---|---|
| | FID↓ | sFID↓ | IS↑ | Precision↑ | Recall↑ | CMMD↓ | Recon↓ | FID↓ | sFID↓ | IS↑ | Precision↑ | Recall↑ | CMMD↓ | Recon↓ |
| RF | 2.76 | 4.05 | 9.51 | 0.70 | 0.59 | 0.0360 | 0.83 | 6.08 | 49.60 | 1.80 | 0.86 | 0.28 | 0.5145 | 17.92 |
| RF+LZN | 2.59 | 3.95 | 9.53 | 0.70 | 0.59 | 0.0355 | 0.41 | 5.68 | 49.32 | 1.96 | 0.87 | 0.30 | 0.3376 | 10.29 |

| Algo. | CelebA-HQ ($256 \times 256$) | | | | | | | LSUN-Bedroom ($256 \times 256$) | | | | | | |
|---|---|---|---|---|---|---|---|---|---|---|---|---|---|---|
| | FID↓ | sFID↓ | IS↑ | Precision↑ | Recall↑ | CMMD↓ | Recon↓ | FID↓ | sFID↓ | IS↑ | Precision↑ | Recall↑ | CMMD↓ | Recon↓ |
| RF | 6.95 | 10.61 | 2.91 | 0.76 | 0.42 | 1.0276 | 26.20 | 6.25 | 16.22 | 2.18 | 0.60 | 0.40 | 0.5218 | 48.72 |
| RF+LZN | 7.17 | 10.33 | 2.92 | 0.76 | 0.45 | 0.4901 | 15.90 | 5.95 | 17.84 | 2.22 | 0.59 | 0.41 | 0.4843 | 37.01 |

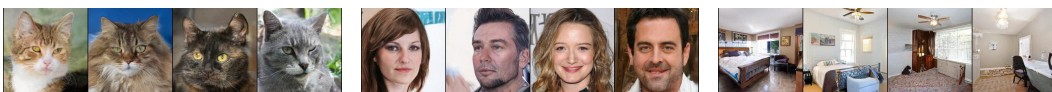

Figure 5: Generated images of RF+LZN on AFHQ-Cat, CelebA-HQ, LSUN-Bedroom. More in App. C.

**Related work.** RCG [44] and Diffusion Autoencoder (DA) [66] also aim to improve image generation using unsupervised representations. RCG leverages a pre-trained representation model, while DA uses an encoder trained jointly with the generator. Similar to LZN, these representations are provided as additional inputs to the generative model. However, because their representation lacks distributional constraints, both methods require training an auxiliary generative model over the representations to enable unconditional generation. In contrast, LZN enforces a Gaussian representation by construction, eliminating this extra step and allowing everything to be trained end-to-end in one stage.

**Results.** We plug LZN latents into the seminal rectified flow (RF) models [53], which is closely related to diffusion models [32, 76, 78]. RF achieved SoTA image generation performance on several datasets [53, 41], and has been used in some latest Stable Diffusion models [23] and AR models [52].

We follow the experimental setup of RF [53], evaluating on four datasets: `CIFAR10`, `AFHQ-Cat`, `CelebA-HQ`, and `LSUN-Bedroom`. The latter three are high-resolution ($256 \times 256$). In addition to standard metrics—FID [31], sFID [58], IS [73], precision [40], recall [40], and CMMD [33]—we also report *reconstruction error* (the $\ell_2$ distance between an image and its reconstruction), which is relevant for applications like image editing via latent manipulation [75, 50]. Results are shown in Tab. 1, with three key findings: **(1)** LZN **latents improve image quality.** During inference, the only difference in RF+LZN is the inclusion of an additional latent drawn from a Gaussian distribution. This improvement indicates that the decoder effectively leverages LZN latents for meaningful generative features. **(2)** LZN **significantly reduces reconstruction error across all datasets,** further confirming that its latents capture essential image information. **(3)** While unconditional generation is important, its image quality often trails that of conditional generation [44]. Compared to RF's `CIFAR10` results in Tab. 4, we find that LZN **substantially reduces the FID gap between conditional and unconditional generation by 59%, and even outperforms conditional RF in sFID and reconstruction error**.

See Fig. 5 for some generated images. Due to space constraints, please refer to App. C for **more details on implementation, datasets, metrics, and additional results such as more generated images and ablation studies**.

# 4 Case Study 2: Unsupervised Representation Learning

**Related work.** Contrastive learning is a popular approach to unsupervised representation learning [10, 11, 28, 12, 14, 26, 13], which pulls similar images (e.g., augmentations of the same image) together in the representation space. A central challenge is avoiding collapse, where all inputs are mapped to the same representation. Common solutions include pushing the representations of dissimilar samples from large batches [10] or memory banks [28] away, or adopting architectural designs that prevent collapse [26, 13, 28]. Other unsupervised representation learning approaches [74, 42] include masked image modeling [43, 4, 27, 63] and training on other auxiliary tasks [84, 59, 65, 25].

**Approach (Fig. 6).** Inspired by contrastive learning, we train an image encoder using Align $(\mathcal{X}, \mathcal{Y})$ (§ 2.2.2), where $\mathcal{X} = \{x_i\}_{i=1}^n$ and $\mathcal{Y} = \{y_i\}_{i=1}^n$ with mapping $k_i = i$ contain image pairs $(x_i, y_i)$ of random augmentations of the same image. Unlike traditional contrastive methods, our approach inherently avoids collapse: different images are mapped to distinct latent zones by design, eliminating the need for large memory banks or specialized architectures. Notably, only a single LZN encoder for

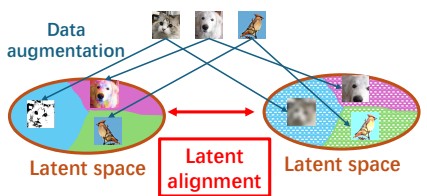

Figure 6: LZN for unsupervised representation learning (§ 4). During training, each image batch undergoes two sets of data augmentations, and latent zones for each set are computed *using the same encoder*. We then apply latent alignment (§ 2.2.2) to train the encoder. At inference, we can use the LZN latents, the encoder outputs (i.e., anchor points), or intermediate encoder outputs (App. D.4). The latter two options avoid the costly latent computation process.

Table 2: Classification accuracy on ImageNet by training a linear classifier on the unsupervised representations. Methods marked with § are based on contrastive learning.[7] The horizontal line separates baselines that perform worse or better than our LZN. All methods use the ResNet-50 architecture [29] for fair comparison.[8]

| Algorithm | Top-1 Acc↑ | Top-5 Acc↑ |
|---|---|---|
| InstDisc§ [85] | 54.0 [28] | - |
| BigBiGAN [21] | 56.6 [28] | - |
| LocalAgg§ [91] | 58.8 [28] | - |
| MoCo§ [28] | 60.2 [10] | - |
| PIRL§ [57] | 63.6 [10] | - |
| CPC v2§ [30] | 63.8 [10] | 85.3 [10] |
| CMC§ [81] | 66.2 [26] | 87.0 [26] |
| SimSiam§[13] | 68.1 [13] | - |
| SimCLR§ [10] | 69.3 [10] | 89.0 [10] |
| MoCo v2§ [12] | 71.7 [12] | - |
| SimCLR v2§ [11] | 71.7 [11] | - |
| BYOL§ [26] | 74.3 [26] | 91.6 [26] |
| DINO§ [9] | 75.3 [9] | - |
| LZN | 69.5 | 89.3 |

Table 3: Classification accuracy on CIFAR10. Baseline results are from [39]. "RF+LZN (no gen)" refers to "RF+LZN" with the RF loss for generation disabled. The horizontal line separates baselines that perform worse or better than our RF+LZN. Note that these results refer to training purely on the CIFAR10 dataset (without pretraining or external data).

| Algorithm | Acc↑ |
|---|---|
| VGG16 | 92.64% |
| ResNet18 | 93.02% |
| ResNet50 | 93.62% |
| ResNet101 | 93.75% |
| RegNetX_200MF | 94.24% |
| RegNetY_400MF | 94.29% |
| MobileNetV2 | 94.43% |
| ResNeXt29(32x4d) | 94.73% |
| ResNeXt29(2x64d) | 94.82% |
| SimpleDLA | 94.89% |
| DenseNet121 | 95.04% |
| PreActResNet18 | 95.11% |
| DPN92 | 95.16% |
| DLA | 95.47% |
| RF+LZN (no gen) | 93.59% |
| RF+LZN | 94.47% |

both $\mathcal{X}$ and $\mathcal{Y}$ is needed and decoders are not required. See App. D.1 for pseudocode of the training process.

**Results.** We follow the canonical setting [10, 26, 28], where LZN is trained on the unlabelled ImageNet dataset using ResNet-50 [29] to learn representations. A *linear* classifier is then trained on top of these representations in a supervised manner, and its accuracy is evaluated on the ImageNet test set. The results are shown in Tab. 2. Note that these results are obtained without any pretraining or use of external data. **We observe that LZN matches or outperforms several established methods in the field, including seminal approaches such as MoCo [28] (LZN outperforms it by 9.3%) and SimCLR [10] (LZN outperforms it by 0.2%).** This is remarkable given that LZN is a new framework capable of supporting not only unsupervised representation learning but also other tasks (§ 3 and 5). However, there remains a significant gap to SoTA performance. **We emphasize that our current results are not fully optimized: (1) Training iteration.** The performance of LZN continues to improve rapidly with ongoing training (App. D), so we expect the gap to SoTA to narrow with full training. **(2) Architecture.** Prior work shows that more advanced architectures like ViT can improve the results significantly [14]. We leave these directions for future work.

Due to space constraints, please refer to App. D for **more details on implementation, datasets, metrics, and additional results such as representation visualization and ablation studies**.

---

[7]Note that we use the term *contrastive learning* broadly to refer not only to methods employing the traditional contrastive loss, but to all approaches that encourage relevant images to share similar representations; see § 4.

[8]It is known that better architectures [11] or training on larger datasets [62] can yield stronger results. To ensure a fair comparison, we include only methods reporting results with *the ResNet-50 architecture* trained on *the* ImageNet *dataset*. This excludes potentially stronger methods lacking ResNet-50 results on ImageNet, such as DINOv2 [62]. See Tab. 6 for additional baselines using other architectures.

Table 4: Conditional image generation quality on `CIFAR10`. The best results are in gray box. Applying LZN to generative models improves or matches RF on all metrics.

| Algo. | FID↓ | sFID↓ | IS↑ | Precision↑ | Recall↑ | CMMD↓ | Recon↓ |
|---|---|---|---|---|---|---|---|
| RF | 2.47 | 4.05 | 9.77 | 0.71 | 0.58 | 0.0253 | 0.69 |
| RF+LZN | 2.40 | 3.99 | 9.88 | 0.71 | 0.58 | 0.0229 | 0.38 |

## 5 Case Study 3: Conditional Generative Modeling and Classification

**Approach (Fig. 3).** Building on § 3, we consider $\mathcal{X} = \{x_i\}_{i=1}^{n}$ and $\mathcal{Y} = \{y_i\}_{i=1}^{m}$, where each $x_i$ is a class label and $y_i$ is an image labeled as $x_{k_i}$. In addition to the image encoder and decoder, we introduce a label encoder-decoder pair. Since labels come from a finite set, both modules share a matrix $A \in \mathbb{R}^{q \times c}$ of label anchor points, where $q$ is the latent dimension and $c$ is the number of classes. The encoder maps a one-hot label $h$ to its anchor via $Ah$. The decoder recovers the class ID of a latent $g$ by first applying FM to obtain its anchor $\mathrm{FM}_A(g; 1)$, and then computing its corresponding class latent zone $\arg\max \mathrm{FM}_A(g; 1)^T A$, where $\mathrm{FM}_A$ denotes FM over anchors in $A$. The training objective extends that of § 3 by adding $\mathrm{Align}(\mathcal{X}, \mathcal{Y})$. After training, the model can perform both conditional and unconditional generation, as well as classification by design. See App. E.1 for detailed pseudocode of the training, generation, and classification processes.

**Related work.** Joint classification and conditional generation are often achieved by augmenting generative models with classification losses or networks [60, 73], treating label inputs to the generator and outputs from the classifier as separate components. In contrast, LZN unifies label inputs and outputs within a shared latent space.

**Results.** Following the setting in § 3, we conduct experiments on `CIFAR10`. Image quality metrics are shown in Tab. 4, and classification accuracies are shown in Tab. 3. Key observations are: **(1)** LZN **improves both image quality and reconstruction error.** Similar to § 3, this confirms that LZN latents capture useful features for generation. **(2)** LZN **achieves classification accuracy on par with SoTA.** Tab. 3 includes SoTA classification accuracy from networks trained solely for classification. The fact that LZN, which jointly performs generation and classification and differs significantly from standard classification pipelines, can match SoTA performance is notable. Currently, LZN lags behind the best `CIFAR10` result by 1%, potentially due to architectural factors: we use the RF encoder (App. E) without classification-specific optimization. With a better architecture design (as in other methods in Tab. 3), LZN could likely improve further. **(3) Joint training on generation and classification improves both.** This is evident from: (i) RF+LZN in Tab. 4 showing better generation quality than in Tab. 1; and (ii) "RF+LZN" achieving higher classification accuracy than "RF+LZN (no gen)" in Tab. 3. These results support our motivation from § 1 that different ML tasks can benefit from each other, and demonstrate that LZN is a promising unified framework for achieving this synergy.

Due to space constraints, please refer to App. E for **more details on implementation, datasets, metrics, and additional results such as generated images and ablation studies**.

## 6 Limitations and Future Work

**(1) Training efficiency.** Training LZN requires backpropagating through the FM trajectory (§ 2.2.1), which is computationally expensive. In App. A, we describe several optimization strategies we implemented to mitigate this cost. To further improve efficiency, we observe an interesting parallel between training LZN and training large language models (LLMs) (see App. G), suggesting that some efficient training techniques developed for LLMs may be applicable here. **(2) Pure generative modeling.** While LZN is fundamentally capable of generative modeling without any auxiliary losses (see App. G), in § 3, we only demonstrate how it can enhance existing generative models. Exploring how to fully leverage LZN for standalone generative modeling remains an open direction for future work. **(3) Improving performance.** Although LZN achieves competitive results in unsupervised representation learning (§ 4) and classification (§ 5), there remains a gap to the SoTA. Bridging this gap is an interesting direction. One promising avenue is to incorporate well-established improvements from the literature that we have not yet adopted, such as more advanced architectural designs, as discussed in § 4 and 5. **(4) Multi-modality and multi-tasks.** In this paper, we focus primarily on image-based applications and at most two tasks simultaneously (§ 5). However, LZN is designed to be extensible: by incorporating additional encoders and decoders, it can naturally support more modalities and perform multiple tasks concurrently (Fig. 1). We leave this exploration to future work.

## Acknowledgement

The authors would like to thank Sangyun Lee for suggesting related work and for helpful discussions, as well as the anonymous reviewers for their valuable feedback. Xuefei Ning acknowledges the support by the National Natural Science Foundation of China (No. 62506197). In addition, the authors gratefully acknowledge Cat Cookie and Cat Doudou for graciously lending their adorable faces for Figs. 2 to 4 and 6.[9]

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

# Appendix Contents

# A    More Details on Latent Computation (§ 2.2.1)

## A.1    The Desired Properties

In this section, we explain in more detail why the construction in § 2.2.1 (approximately) satisfies the two desired properties.

- **Prior distribution is Gaussian.** By definition, the distribution of latent $z \sim$ Uniform $\{z_1, \ldots, z_n\}$ is induced by (1) drawing $a \sim$ Uniform $\{a_1, \ldots, a_n\}$ and $\epsilon \sim \mathcal{N}(0, \mathbf{I})$, (2) computing $s_{1-g} = (1-g)a + g\epsilon$, and (3) computing $\text{IFM}_x(s_{1-g}; 0)$. In the above process, $s_{1-g} \sim \pi_{1-g}$ by definition. Due to the property of FM discussed in § 2.2.1, $\text{IFM}_x(s_{1-g}; 0) \sim \pi_0$ when $s_{1-g} \sim \pi_{1-g}$. Therefore, the latent $z \sim \pi_0 = \mathcal{N}(0, \mathbf{I})$.
- **Disjoint latent zones.** Each latent point $z \sim \mathcal{N}(0, \mathbf{I})$ can be uniquely map to one of $\{a_1, \ldots, a_n\}$ through the defined FM. We define the latent zone of $i$-th sample as the set of latents that map to $a_i$: $Z_i = \{z : \text{FM}_x(z; 1) = a_i\}$. The probability that the latent computed through Eq. (1) falls in the incorrect latent zone can then be defined as $\mathbb{P}\left(\text{IFM}_x(a_i, \epsilon; 0) \in Z_j\right)$ for $i \neq j$, where the probability is over the randomness of $\epsilon \sim \mathcal{N}(0, \mathbf{I})$. We can see that this probability can be made arbitrarily small by choosing a sufficiently small $g$. This is because that to make the latent fall in the incorrect latent zone, we need to have $\|\epsilon\|$ on the scale of $\left\|\frac{(1-g)(a_i-a_j)}{g}\right\|$, whose probability $\rightarrow 0$ when $g \rightarrow 0$. To make this intuition more precise, we give the closed form of this probability for a toy one-dimensional case below.

**Theorem 1.** *Assume that there are $n = 2$ samples $x_1, x_2$, with their anchor points $a_1 = -1, a_2 = 1$. We have*

$$\mathbb{P}\left(\text{IFM}_x(a_1, \epsilon; 0) \in Z_2\right) = \mathbb{P}\left(\text{IFM}_x(a_2, \epsilon; 0) \in Z_1\right) = \Phi\left(\frac{g-1}{g}\right), \tag{3}$$

*where $\Phi$ is the CDF function of the standard Gaussian distribution.*

*Proof.* With a slight abuse of notation, we define $\text{FM}_x(s; t_1, t_2) = s + \int_{\tau=t_1}^{t_2} V(s_\tau, \tau)\mathrm{d}\tau$ as following the FM trajectory from $t_1$ to $t_2$. We generalize the latent zone definition above to all time steps as the latents at time step $t$ that map to the anchor point $a_i$: $Z_i^t = \{z : \text{FM}_x(z; t, 1) = a_i\}$. Due to symmetry, we know that $Z_1^t = (-\infty, 0)$ and $Z_2^t = (0, \infty)$. Therefore, we have

$$\mathbb{P}\left(\text{IFM}_x(a_1, \epsilon; 0) \in Z_2\right) = \mathbb{P}\left(-(1-g) + g\epsilon > 0\right) = \mathbb{P}\left(\epsilon > \frac{1-g}{g}\right) = \Phi\left(\frac{g-1}{g}\right),$$

where $\Phi(\cdot)$ is the CDF function of Gaussian distribution. Similarly, we can get that $\mathbb{P}\left(\text{IFM}_x(a_2, \epsilon; 0) \in Z_1\right) = \Phi\left(\frac{g-1}{g}\right)$. $\qquad\square$

## A.2    More Implementation Details

We find that in the training or inference of some tasks benefit from using a more concentrated latent distribution:

$$z_i = C(x_1, \ldots, x_n)_i \triangleq \text{IFM}_x(a_i, \alpha\epsilon_i; 0), \tag{4}$$

where $\epsilon_i \sim \mathcal{N}(0, \mathbf{I})$ and $\alpha \in [0, 1)$ is the scaling factor. Similar techniques have been used in prior generative models for improving sample quality [35, 37, 6, 82].

## A.3    Efficiency Optimizations

We introduce a series of efficiency optimization techniques so that the training of LZN can scale up to large models and large batch sizes.

**Minibatch approximation (reducing memory and computation cost).** By design, latent computation requires using *all* samples, which is infeasible for large datasets. In practice, we approximate this by using only the current minibatch as $x_1, \ldots, x_n$, which significantly reduces memory and computation cost.

Note that this approximation has nuanced implications on the two desired properties discussed in § 2.2.1.

- **Minibatch approximation still preserves the Gaussian prior.** LZN ensures that the latent distribution within each minibatch is approximately $\mathcal{N}(0, \mathbf{I})$. As a result, the overall latent distribution becomes a mixture of Gaussians with the same parameters, which is still $\mathcal{N}(0, \mathbf{I})$. Therefore, the global prior remains valid under the minibatch approximation.
- **However, minibatch approximation violates the disjoint zone property.** This is because the latent zones of each sample now depends on other samples in the same batch, which could change across different batches. Despite this approximation, our experiments show it performs well.
  - In generative modeling (§ 3 and 5), the latent does not need perfect zone disjointness—as long as it provides some information about the input sample, it can help reduce the variance needed to learn by the generative model (rectified flow in our case) and improve the generation quality.
  - In representation learning (§ 4), latent alignment occurs *within a single batch*. Thus, inconsistency across batches is irrelevant.
  - In classification (§ 5), we only need to map samples *within a single batch* to the latent zones of labels. Thus, inconsistency across batches is irrelevant.

  That said, *larger batch sizes* can improve the accuracy of latent zones and thus improve performance (§ 5).

**Custom gradient checkpointing (reducing memory cost).** In PyTorch, forward passes store intermediate results for use in backpropagation, incurring significant memory cost. Gradient checkpointing[10] reduces memory usage (with the cost of extra computation) by selectively discarding intermediates in the forward pass and recomputing them during the backward pass. This technique is typically applied within neural networks. In our case, we discover that the main memory bottleneck lies in latent computation, which has memory complexity $\mathcal{O}(n^2 qr)$, where $n$ is the number of samples, $q$ the latent dimension, and $r$ the solver steps. We design a custom strategy that skips storing velocity computations and retains only the latent trajectories $s_t$. This reduces memory complexity to $\mathcal{O}(nqr)$, which makes the training far more manageable.

**Latent parallelism (making training scalable with multi-GPU).** For the same reason discussed above, the main computation overhead also lies in latent computation. A natural idea is to parallelize it with multi-GPU. We partition the data samples across GPUs, and each GPU computes anchor points for its assigned subset. These anchor points are then broadcast to all GPUs, allowing each to compute latents for its own samples using the complete set of anchors. To ensure that gradients can propagate back correctly through the anchor points to the originating GPUs, we use the undocumented PyTorch function `torch.distributed.nn.functional.all_gather`, which—unlike the standard `torch.distributed.all_gather`—maintains gradient flow to the original sources.

# B    More Details on Latent Alignment (§ 2.2.2)

## B.1    More Implementation Details

Optionally, we can apply a logarithm to the assignment probability to make the loss resemble a standard cross-entropy formulation. In that case, our proposed alignment objective is:

$$\text{Align}(\mathcal{X}, \mathcal{Y}) \triangleq \max \sum_{i=1}^{m} \max_{t \in \{t_u, \dots, t_r\}} \log \mathbb{P}\left(a_{k_i} | s_t^i\right). \tag{5}$$

## B.2    Efficiency Optimizations

We apply the same efficiency optimizations in App. A.3 in latent alignment.

# C    More Details and Results on Case Study 1

## C.1    Algorithm Pseudocode

Fig. 7 and Fig. 8 show side-by-side comparisons of the training and generation processes of RF and RF+LZN.

---

[10] https://pytorch.org/docs/stable/checkpoint.html

**Algorithm 1:** RF training

**Input** : Training set: $\mathcal{X}$
     Decoder: $D_x$
     Number of iterations: $T$
     Batch size: $B$

1 **for** *iteration* $\leftarrow 1, \ldots, T$ **do**
2   $x_1, \ldots, x_B \leftarrow$ Draw samples from $\mathcal{X}$
3   $\epsilon_1, \ldots, \epsilon_B \leftarrow$ Gaussian noise
4   $t_1, \ldots, t_B \leftarrow$ Random RF timesteps
5   $\xi_i \leftarrow (1 - t_i)\epsilon_i + t_i x_i$
6   Training using $D_x(\xi_i)$

**Algorithm 2:** RF+LZN training

**Input** : Training set: $\mathcal{X}$
     Decoder: $D_x$
     Encoder: $E_x$ (used by $C$)
     Number of iterations: $T$
     Batch size: $B$

1 **for** *iteration* $\leftarrow 1, \ldots, T$ **do**
2   $x_1, \ldots, x_B \leftarrow$ Draw samples from $\mathcal{X}$
3   $z_1, \ldots, z_B \leftarrow C(x_1, \ldots, x_B)$
4   $\epsilon_1, \ldots, \epsilon_B \leftarrow$ Gaussian noise
5   $t_1, \ldots, t_B \leftarrow$ Random RF timesteps
6   $\xi_i \leftarrow (1 - t_i)\epsilon_i + t_i x_i$
7   Training using $D_x(\xi_i; z_i)$

Figure 7: **Comparison between the training processes of RF and RF+LZN. Left:** A simplified illustration of the standard RF [53] training process. In each iteration, a batch of real samples and a batch of Gaussian noise are drawn and interpolated to produce noisy inputs, which are then passed through the decoder network to compute the loss. **Right:** A simplified illustration of the RF+LZN training process. The key differences are highlighted in gray: we compute LZN latents for the samples using the method in § 2.2.1, and provide these latents as an additional input to the RF decoder.

**Algorithm 3:** RF generation

**Input** : Decoder: $D_x$

1 $\xi \leftarrow$ Gaussian noise
2 Generated sample $\leftarrow D_x(\xi)$

**Algorithm 4:** RF+LZN generation

**Input** : Decoder: $D_x$

1 $\xi \leftarrow$ Gaussian noise
2 $z \leftarrow$ Gaussian noise
3 Generated sample $\leftarrow D_x(\xi; z)$

Figure 8: **Comparison between the generation processes of RF and RF+LZN. Left:** A simple illustration of the standard RF generation process [53]. The decoder takes Gaussian noise as input and generates a sample. The actual process is iterative, but we leave out the steps for simplicity and only show the starting input (Gaussian noise) and the final output (the generated image). **Right:** A simple illustration of the RF+LZN generation process. The main differences are shown in gray: we sample extra LZN latents from Gaussian noise and use them as additional inputs to the RF decoder during the iterative generation process.

## C.2 More Implementation Details

**Architecture.**

- **Decoder.** The only change to the RF architecture [53] is concatenating the LZN latent with the timestep embedding.
- **Encoder.** We extend the UNet encoder in RF [53] by connecting the output of each ResNet block with a latent transformation block. The sum of the outputs of the latent transformation blocks forms the LZN latent. Each latent transformation block consists of: (1) a $1 \times 1$ convolution that projects the ResNet output to 20 channels, reducing dimensionality; and (2) a small MLP with a 200-dimensional hidden layer that outputs the latent from the flattened convolution output.

## C.3 More Experimental Settings

**Datasets.**

- CIFAR10 ($32 \times 32$) [38] contains 50000 training images and 10000 test images of 10 classes of objects. We only utilize the training set for this experiment.
- AFHQ-Cat ($256 \times 256$) [15] contains 5153 catimages.

- `CelebA-HQ` ($256 \times 256$) **[34]** contains 30000 face images.
- `LSUN-Bedroom` ($256 \times 256$) **[86]** contains 3033042 bedroom images.

**Metrics.**

- **FID [31] and sFID [58]** evaluate the similarity between real and generated images by projecting both into the latent space of a pretrained network (e.g., Inception-v3 [80]), fitting each set of latents with Gaussian distributions, and computing their Wasserstein-2 distance. The key difference between FID and sFID is the feature layer used: FID uses pooled features, while sFID uses intermediate features, making it more sensitive to spatial details.
- **Inception score (IS) [73]** measures image quality by assessing both the quality of each image (how confidently a classifier predicts a class) and diversity across all images (coverage over different classes). Since the classifier is trained on `ImageNet` [17], IS is best suited for natural image datasets like `CIFAR10`. We report IS for all datasets for completeness.
- **Precision and recall [40]** evaluate the quality and coverage of generated images. Intuitively, precision measures the fraction of generated images that are close to real ones, while recall measures the fraction of real images that are close to the generated ones.
- **CMMD [33]** measure the MMD distances between the CLIP embeddings of the real images and generated images. Compared to FID, it is reported to align better with human preference and have better sample efficiency.
- **Reconstruction error** measures how well a generative model can reconstruct an input image. This reflects the model's representational power and is crucial for applications like image editing, where edits are made by modifying the image's latent representation []. For RF, we first apply the inverse ODE to map the image to its latent representation, then use the forward ODE to reconstruct the image, and compute the $\ell_2$ distance between the original and reconstructed images. For RF+LZN, we add an initial step: compute the image's LZN latent $C(\mathcal{X})$ and feed it into the RF latent computation process as an additional input.

Following the convention [53, 77, 48, 90, 49], the metrics are all computed using the training set of the dataset.

For FID, sFID, IS, precision, recall, and CMMD, we subsample the training set and generate the same number of samples to compute the metrics. The number of samples are:

- `CIFAR10`: 50000 (the whole training set).
- `AFHQ-Cat`: 5120, the largest multiple of the batch size (256) that is less than or equal to the training set size (5153).
- `CelebA-HQ`: 29952, the largest multiple of the batch size (256) that is less than or equal to the training set size (30000).
- `LSUN-Bedroom`: 29952, the largest multiple of the batch size (256) that is less than or equal to 30000. We limit the number of samples to 30000 so that the computation cost of the metrics are reasonable.

For reconstruction error, we randomly sample a batch of images (2000 for `CIFAR10` and 256 for the other datasets) from the training set. Each image is reconstructed 20 times (note that the LZN latents $C(\mathcal{X})$ have randomness). We report the average metric over all reconstructions.

Note that for all the random subsampling procedures mentioned above, we ensure the sampled sets are consistent between RF and RF+LZN, so that the resulting metrics are directly comparable.

**Sampler.** RF requires a sampler to numerically solve the ODE (integral) trajectory for sample generation. For both RF and RF+LZN, we use the RK45 sampler from RF [53], which adaptively determines the number of steps. In App. C.4, we also analyze the effect of varying the number of sampling steps using the Euler sampler [].

**Hyperparameters.**

- `CIFAR10`
  - RF:
  - Batch size: 2000
  - Optimizer: Adam
  - Decoder learning rate: 0.001

- Gradient clipping: 1.0
- Number of parameters in decoder: 61804419
  - RF+LZN:
- Batch size: 2000
- Optimizer: Adam
- Decoder learning rate: 0.001
- Encoder learning rate: 0.000025
- Gradient clipping: 1.0
- Latent dimension: 200
- Number of parameters in decoder: 61906819
- Number of parameters in encoder: 49790260
- AFHQ-Cat
  - RF:
- Batch size: 256
- Optimizer: Adam
- Decoder learning rate: 0.0002
- Gradient clipping: 1.0
- Number of parameters in decoder: 65574549
  - RF+LZN:
- Batch size: 256
- Optimizer: Adam
- Decoder learning rate: 0.0002
- Encoder learning rate: 0.000002
- Gradient clipping: 1.0
- Latent dimension: 200
- Number of parameters in decoder: 65676949
- Number of parameters in encoder: 87768896
- CelebA-HQ
  - RF:
- Batch size: 256
- Optimizer: Adam
- Decoder learning rate: 0.0002
- Gradient clipping: 1.0
- Number of parameters in decoder: 65574549
  - RF+LZN:
- Batch size: 256
- Optimizer: Adam
- Decoder learning rate: 0.0002
- Encoder learning rate: 0.000004
- Gradient clipping: 1.0
- Latent dimension: 200
- Number of parameters in decoder: 65676949
- Number of parameters in encoder: 87768896
- LSUN-Bedroom
  - RF:
- Batch size: 256
- Optimizer: Adam
- Decoder learning rate: 0.0002
- Gradient clipping: 1.0
- Number of parameters in decoder: 65574549
  - RF+LZN:
- Batch size: 256

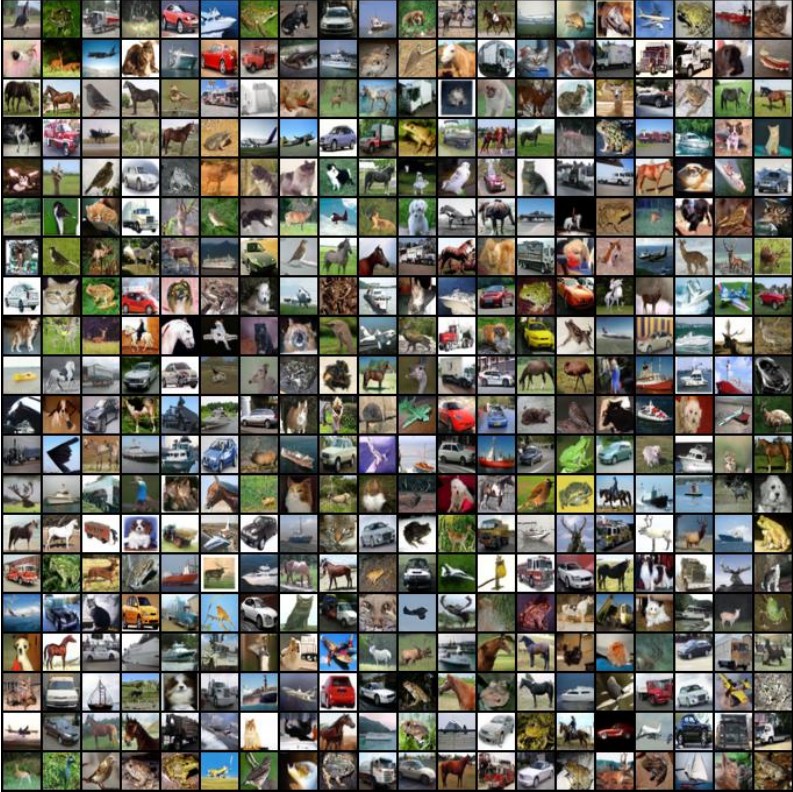

Figure 9: Generated images of RF on `CIFAR10`.

- Optimizer: Adam
- Decoder learning rate: 0.0002
- Encoder learning rate: 0.000002
- Gradient clipping: 1.0
- Latent dimension: 200
- Number of parameters in decoder: 65676949
- Number of parameters in encoder: 87768896

**Computation cost.** Excluding the computation cost of periodic evaluation (i.e., only counting the computation cost of model training), each RF+LZN experiment takes:

- `CIFAR10`: 8 hours on 16 A100 (40 GB) GPUs.
- `AFHQ-Cat`: 10 hours on 32 A100 (40 GB) GPUs.
- `CelebA-HQ`: 58 hours on 32 A100 (40 GB) GPUs.
- `LSUN-Bedroom`: 341 hours on 32 A100 (40 GB) GPUs.

### C.4 More Results

**Generated images.** The generated images of RF and RF+LZN are in Figs. 9 to 16.

**Ablation studies on FID implementation.** It is known that subtle differences in FID implementation can result in different results [64]. In our main experiments, we use the implementation in consistency models [77]. In Tab. 5, we additionally show the FID using two other implementations: RF [53] and clean FID [64]. We can see that, while the numbers are different, the relative ranking across all three implementations is consistent. Especially, RF+LZN achieves the best FID in three out of four datasets.

**Ablation studies on sampling steps.** In this experiment, we use the Euler sampler with varying numbers of sampling steps. As shown in Fig. 17, RF+LZN generally achieves better FID than the RF baseline across most settings. Notably, in the only case where RF+LZN performs worse than RF in

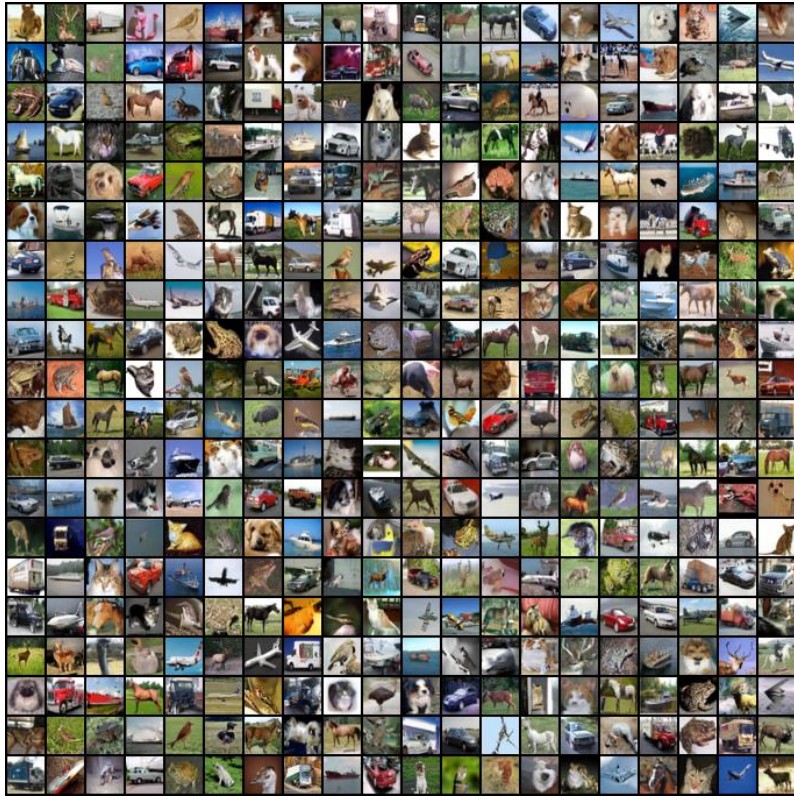

Figure 10: Generated images of RF+LZN on CIFAR10.

Table 5: FID with different implementations for unconditional image generation. "CM" denotes consistency models [77]; "RF" denotes Rectified Flow [53]; "clean" denotes clean FID [64]. The best results are in gray box .

| Algo. | CIFAR10 ($32 \times 32$) | | | AFHQ-Cat ($256 \times 256$) | | |
|---|---|---|---|---|---|---|
| | FID (clean)↓ | FID (RF)↓ | FID (CM)↓ | FID (clean)↓ | FID (RF)↓ | FID (CM)↓ |
| RF | 3.18 | 2.77 | 2.76 | 5.99 | 6.20 | 6.08 |
| RF+LZN | 3.05 | 2.61 | 2.59 | 5.66 | 5.69 | 5.68 |

| Algo. | CelebA-HQ ($256 \times 256$) | | | LSUN-Bedroom ($256 \times 256$) | | |
|---|---|---|---|---|---|---|
| | FID (clean)↓ | FID (RF)↓ | FID (CM)↓ | FID (clean)↓ | FID (RF)↓ | FID (CM)↓ |
| RF | 7.10 | 7.00 | 6.95 | 6.39 | 6.25 | 6.25 |
| RF+LZN | 7.31 | 7.23 | 7.17 | 5.88 | 5.87 | 5.95 |

Tab. 1, we observe that the underperformance occurs only at the highest number of sampling steps in the Euler sampler (Fig. 17c).

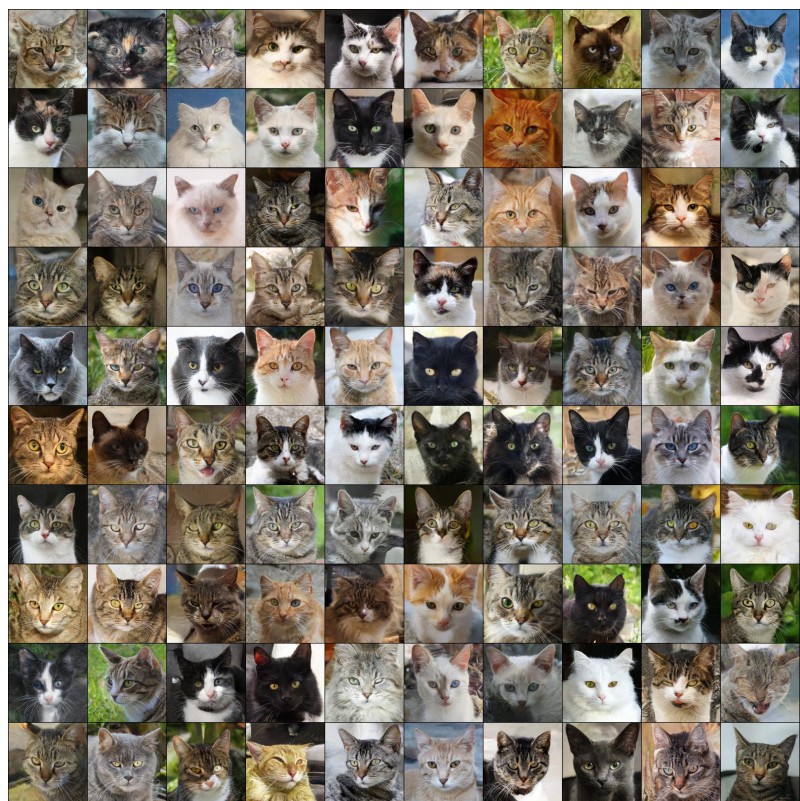

Figure 11: Generated images of RF on `AFHQ-Cat`.

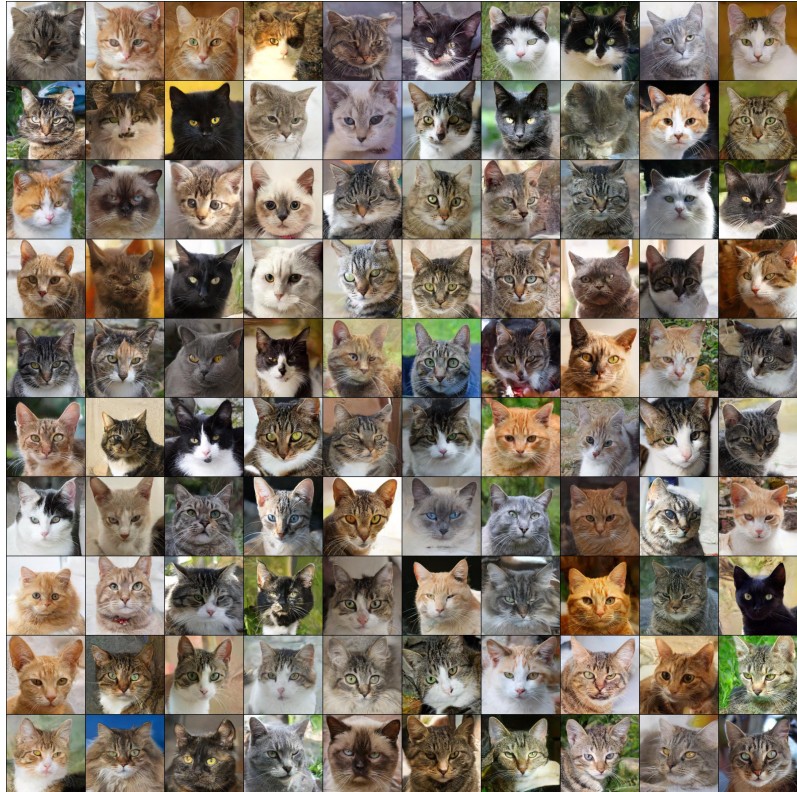

Figure 12: Generated images of RF+LZN on `AFHQ-Cat`.

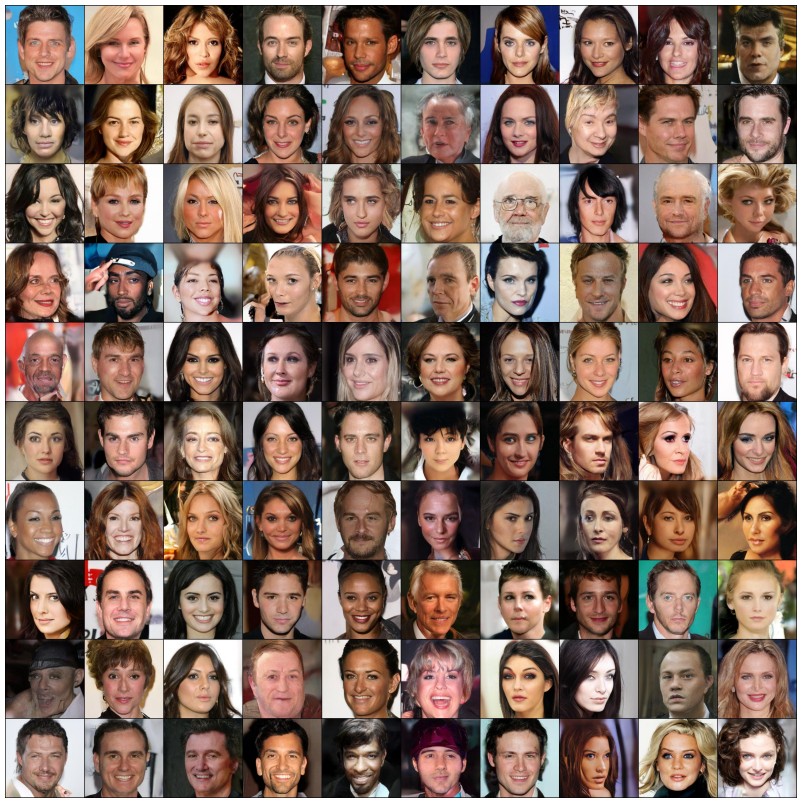

Figure 13: Generated images of RF on `CelebA-HQ`.

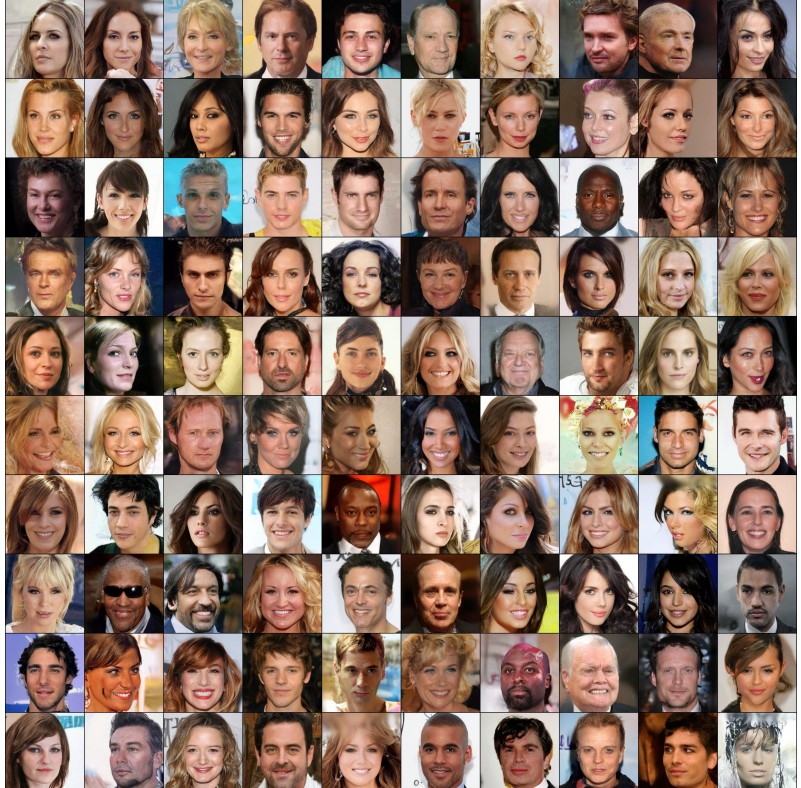

Figure 14: Generated images of RF+LZN on `CelebA-HQ`.

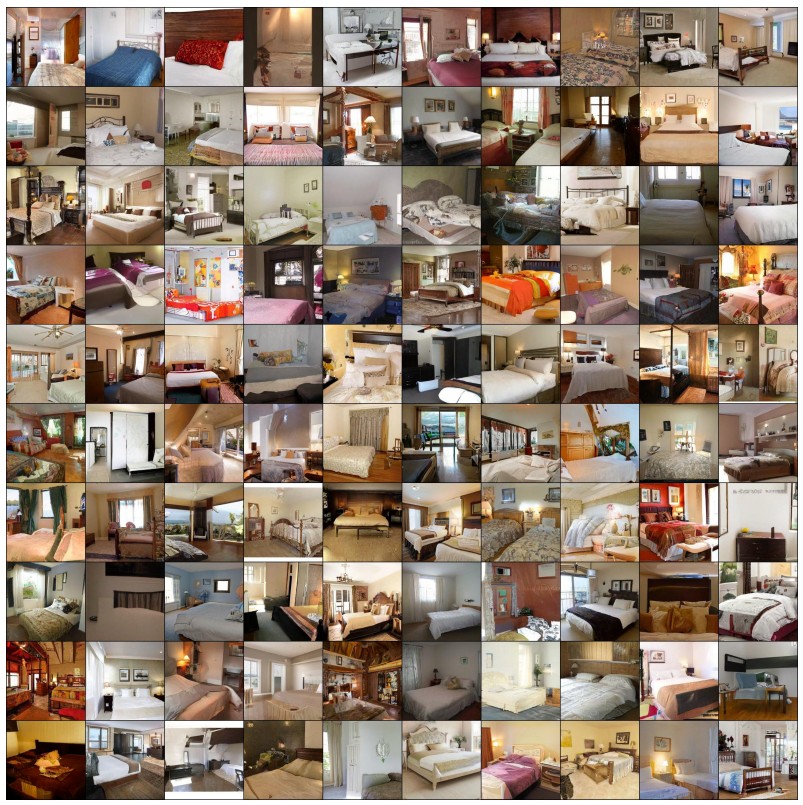

Figure 15: Generated images of RF on LSUN-Bedroom.

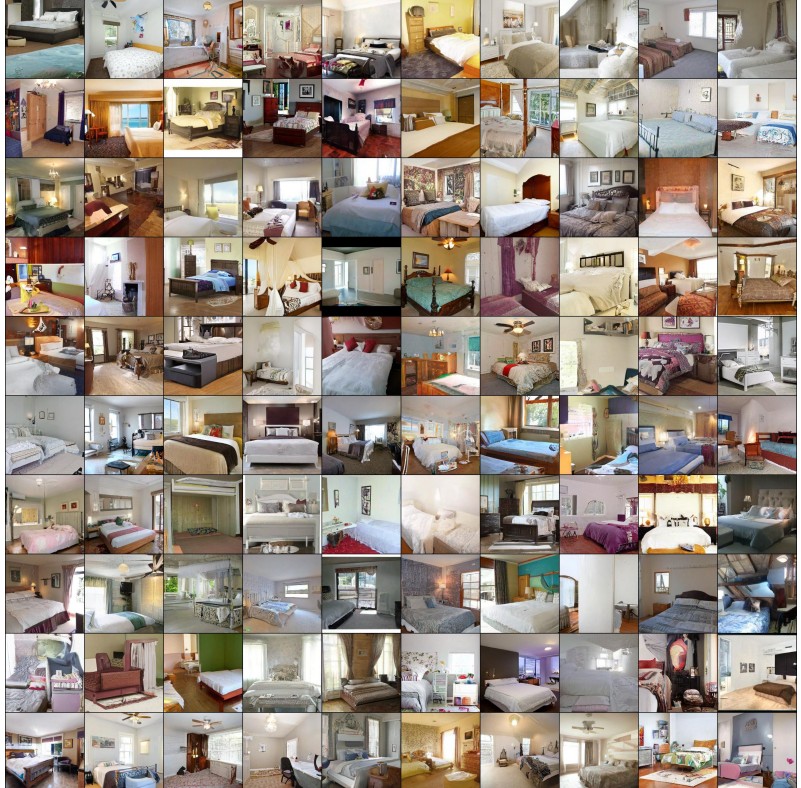

Figure 16: Generated images of RF+LZN on LSUN-Bedroom.

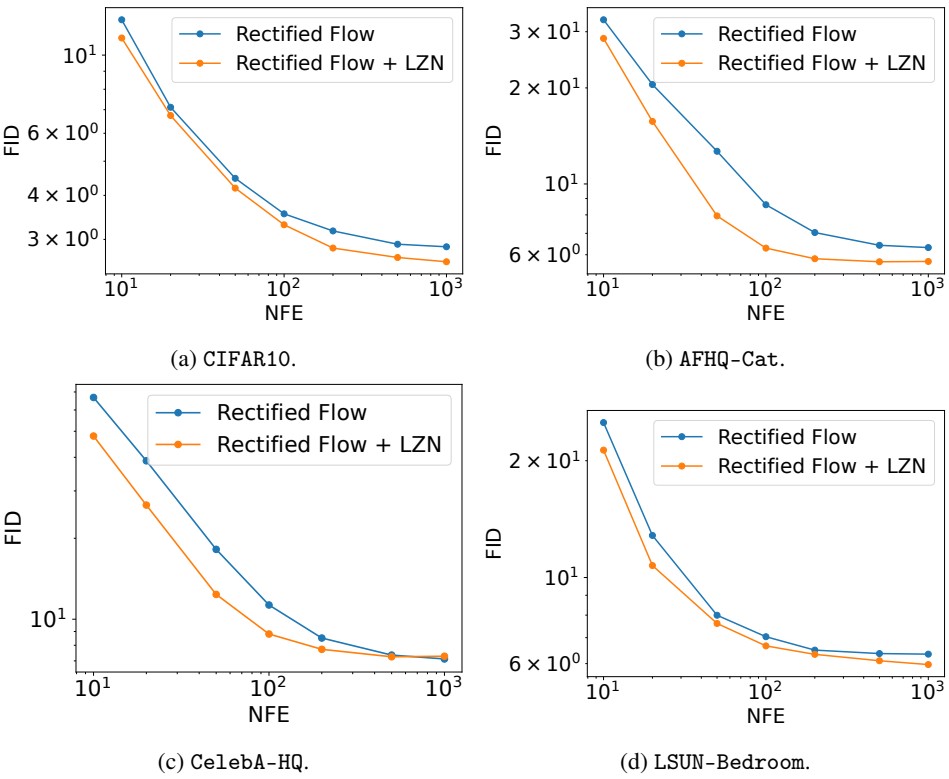

(a) `CIFAR10`.

(b) `AFHQ-Cat`.

(c) `CelebA-HQ`.

(d) `LSUN-Bedroom`.

Figure 17: FID vs. number of sampling steps in the Euler sampler. RF+LZN outperforms RF in most cases.

# D  More Details and Results on Case Study 2

## D.1  Algorithm Pseudocode

Alg. 5 shows the pseudocode of the training process.

After training, the encoder $E_x$ can be used to obtain image representations. We provide several strategies for extracting these representations. Please see App. D.4 for details.

## D.2  More Implementation Details

**Architecture.** To remain consistent with prior work [28, 10, 26], we use the ResNet-50 architecture [29] as the encoder for LZN. The only modification we make is replacing all batch normalization layers with group normalization. However, our early experiments indicate that this change does not lead to significant performance differences.

For the projection head following the ResNet-50 output, we use an MLP with one hidden layer, as in [10, 11].

**Data augmentation.** We follow the same data augmentation strategy as in [11].

**Representation.** Following prior work [10, 11], after training the ResNet-50, we discard the projection head and use only the ResNet-50 backbone to extract representations for training the linear classifier. As a result, obtaining representations from LZN in this way does not require going through the LZN latent computation process $C$, and thus has the same computational efficiency as baseline methods.

**Objective.** We use the version with log (Eq. (5)).

## D.3  More Experimental Settings

**Datasets.** We use the `ImageNet` dataset, which contains 1281167 training images and 50000 validation images. LZN is trained on the training set, and classification accuracy is evaluated on the validation set.

**Hyperparameters.**
- Batch size: 8192
- Optimizer: Adam
- Learning rate: 8e-4
- Gradient clipping: 1.0
- Latent dimension: 256
- Number of parameters: 24032832
- $\alpha$: 0.45

**Computation cost.** Excluding the computation cost of periodic evaluation (i.e., only counting the computation cost of model training), each LZN experiment takes 1800 hours on 128 A100 (40 GB) GPUs.

## D.4  More Results

**More baselines.** Tab. 6 shows the result with more baselines that are not using the ResNet-50 architecture.

**Visualizing the learned representations.** We take images from randomly selected 20 classes from the validation set of ImageNet and computed their embeddings using the trained LZN model. We chose the validation set to ensure that the results are not influenced by training set overfitting. We then projected these embeddings into a 2D space using t-SNE–a widely used method for visualizing high-dimensional representations, following seminal works such as SimCLR [10]. The resulting

---

[11]Note that we use the term *contrastive learning* broadly to refer not only to methods employing the traditional contrastive loss, but to all approaches that encourage relevant images to share similar representations; see § 4.

**Algorithm 5:** Unsupervised representation learning with LZN

**Input** : Training set: $\mathcal{X}$
Encoder: $E_x$
Number of iterations: $T$
Batch size: $B$

**1 for** *iteration* $\leftarrow 1, \dots, T$ **do**
**2**     $x_1, \dots, x_B \leftarrow$ Draw samples from $\mathcal{X}$
**3**     $x_i', x_i'' \leftarrow$ Two random augmentations of $x_i$
**4**     Training $E_x$ using Align $\left(\{x_1', \dots, x_B'\}, \{x_1'', \dots, x_B''\}\right)$

Table 6: Classification accuracy on ImageNet by training a linear classifier on the unsupervised representations. Methods with $\S$ are based on contrastive learning.[11] The horizontal line separates baselines that perform worse or better than our LZN. "R" means "ResNet".

| Algorithm | Architecture | Top-1 Acc↑ | Top-5 Acc↑ |
|---|---|---|---|
| Colorization [88] | R101 | 39.6 [28] | - |
| Jigsaw [59] | R50w2× | 44.6 [28] | - |
| Exemplar [22] | R50w3× | 46.0 [28] | - |
| DeepCluster [8] | VGG | 48.4 [28] | - |
| CPC v1$^\S$ [61] | R101 | 48.7 [28] | - |
| RelativePosition [20] | R50w2× | 51.4 [28] | - |
| InstDisc$^\S$ [85] | R50 | 54.0 [28] | - |
| Rotation [25] | Rv50w4× | 55.4 [28] | - |
| BigBiGAN [21] | R50 | 56.6 [28] | - |
| LocalAgg$^\S$ [91] | R50 | 58.8 [28] | - |
| MoCo$^\S$ [28] | R50 | 60.2 [10] | - |
| BigBiGAN [21] | Rv50w4× | 61.3 [28] | 81.9 [10] |
| PIRL$^\S$ [57] | R50 | 63.6 [10] | - |
| CPC v2$^\S$ [30] | R50 | 63.8 [10] | 85.3 [10] |
| CMC$^\S$ [81] | R50 | 66.2 [26] | 87.0 [26] |
| SimSiam$^\S$[13] | R50 | 68.1 [13] | - |
| SimCLR$^\S$ [10] | R50 | 69.3 [10] | 89.0 [10] |
| MoCo v2$^\S$ [12] | R50 | 71.7 [12] | - |
| SimCLR v2$^\S$ [11] | R50 | 71.7 [11] | - |
| BYOL$^\S$ [26] | R50 | 74.3 [26] | 91.6 [26] |
| DINO$^\S$ [9] | R50 | 75.3 [9] | - |
| DINO$^\S$ [9] | ViT-S | 77.0 [9] | - |
| DINO$^\S$ [9] | ViT-B/16 | 78.2 [9] | - |
| DINO$^\S$ [9] | ViT-S/8 | 79.7 [9] | - |
| DINO$^\S$ [9] | ViT-B/8 | 80.1 [9] | - |
| I-JPEA [3] | ViT-B/16 | 72.9 [3] | - |
| I-JPEA [3] | ViT-L/16 | 77.5 [3] | - |
| I-JPEA [3] | ViT-H/14 | 79.3 [3] | - |
| I-JPEA [3] | ViT-H/16$_{448}$ | 81.1 [3] | - |
| LZN | R50 | 69.5 | 89.3 |

t-SNE plot is in Fig. 18. We can see that the samples from different classes are well-clustered. This suggests LZN learns meaningful image representations.

**Ablation studies on representation choice.** Prior work [10, 11] has shown that the choice of feature extraction layer significantly affects downstream performance. In particular, removing the projection head often improves results. Motivated by this, we explore various feature extraction strategies for LZN, which offers more flexibility due to its unique latent computation process (§ 2.2.1). Specifically, we compare the following methods:

- **With latent.** Use the latent representation from LZN (see § 2.2.1) to train the classifier.
- **With latent ($\alpha = 0$).** Same as above but with $\alpha = 0$ in the latent computation.
- **With head.** Use the anchor point (i.e., encoder output before FM computation).
- **Without head.** Use the ResNet backbone output (before the projection head).

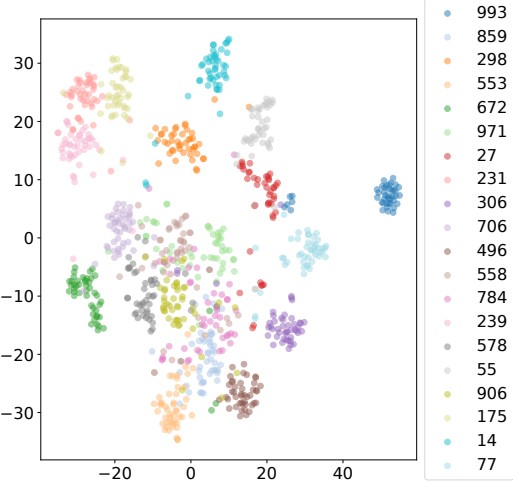

Figure 18: t-SNE visualization of LZN representations projected into 2D for 20 randomly selected ImageNet validation classes. Images from the same class form distinct clusters, indicating that LZN learns meaningful image representations.

The first two methods are specific to LZN, while the last two follow the design commonly used in prior contrastive learning work [10, 11].

The results are shown in Fig. 19. We observe the following:

- LZN latent achieves the lowest prediction accuracy. As discussed in App. A.2, the LZN latents are reliable only when computed over the full dataset. However, for efficiency, both training and inference rely on minibatches to approximate the latent representations. This approximation increases the size of the latent zones, leading to potential overlap between the zones of different samples across batches, which inevitably degrades downstream classification performance.
- In comparison, LZN with $\alpha = 0$ yields significantly higher accuracy. This improvement can be attributed to the reduced likelihood of overlap between latent zones when $\alpha = 0$, making the resulting representations more distinct and less noisy.
- The final two methods, "with head" and "without head", do not involve latent computation and are therefore more efficient. Consistent with findings from prior contrastive learning studies [10], we observe that "without head" performs substantially better. As explained in [10], the projection head often discards important information—such as types of data augmentation—in order to minimize the training loss. In contrast, layers preceding the head might retain richer and more discriminative features, which are more useful for downstream classification tasks.

**Ablation studies on the number of training steps.** Fig. 20 shows classification accuracy over training iterations. Accuracy continues to improve rapidly at the end of training, suggesting that with more training, the gap between LZN and the SoTA could be further reduced.

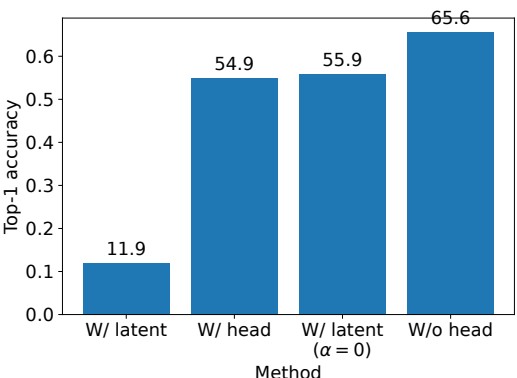

Figure 19: LZN's linear classification accuracy with different feature extraction methods. Note that this experiment uses fewer iterations (1060000) than the main experiment (5000000) and omits data augmentation when training the linear classifier (used in the main experiment), so the accuracies are lower than the main experiment.

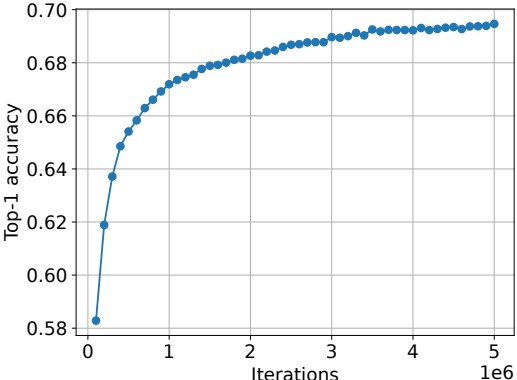

Figure 20: LZN's linear classification accuracy vs. training iteration. The accuracy is still improving at a fast rate at the end of training. More training might further improve the result.

# E More Details and Results on Case Study 3

## E.1 Algorithm Pseudocode

Alg. 6, Fig. 21, and Alg. 9 show the algorithm pseudocode of the training, generation, and classification process.

## E.2 More Implementation Details

**Architecture.**

- **Image decoder.** For RF, we modify the original architecture [53] to include a one-hot encoding of the class label as an additional input, concatenated with the timestep embedding. For RF+LZN, we apply the same modification on top of the architecture described in App. C. For unconditional generation, this one-hot encoding is deterministically derived from the LZN latent: given a LZN latent, we use the class label decoder to predict the class and then encode it as a one-hot vector. As a result, the decoder's output remains fully determined by the LZN and RF latents, consistent with § 3. For conditional generation, this one-hot encoding is given as a condition, and LZN latents are sampled from the corresponding latent zone (as described in § 2.2.1).
- **Image encoder.** Same as that of App. C.

**Label FM.** The method in § 2.2.2 implicitly assumes a uniform distribution over class labels. However, due to sampling randomness during training, each batch may have an imbalanced class

---

**Algorithm 6:** RF+LZN training (with class labels)

---

**Input** : Training set: labels $\mathcal{X}$ and images $\mathcal{Y}$
           Image decoder: $D_y$
           Image encoder: $E_y$ (used by $C$)
           Label anchors: $A$ (used by Align)
           Number of iterations: $T$
           Batch size: $B$

**1 for** *iteration* $\leftarrow 1, \ldots, T$ **do**
**2**     $y_1, \ldots, y_B \leftarrow$ Draw images from $\mathcal{Y}$
**3**     $z_1, \ldots, z_B \leftarrow C(y_1, \ldots, y_B)$
**4**     $\epsilon_1, \ldots, \epsilon_B \leftarrow$ Gaussian noise
**5**     $t_1, \ldots, t_B \leftarrow$ Random RF timesteps
**6**     $\xi_i \leftarrow (1 - t_i)\epsilon_i + t_i y_i$
**7**     Training using $\text{Align}(\mathcal{X}, \{y_1, \ldots, y_B\})$ and RF loss on $D_y(\xi_i; z_i)$ (a weighted loss between the two)

---

---

**Algorithm 7:** RF+LZN generation (unconditional)

---

**Input** : Image decoder: $D_y$

**1** $\xi \leftarrow$ Gaussian noise
**2** $z \leftarrow$ Gaussian noise
**3** Generated sample $\leftarrow D_y(\xi; z)$

---

---

**Algorithm 8:** RF+LZN generation (unconditional)

---

**Input** : Image decoder: $D_y$
           Label set: $c_1, \ldots, c_n$
           Class ID: $k$ (i.e., the class is $c_k$)

**1** $\xi \leftarrow$ Gaussian noise
**2** $z \leftarrow C(\{c_1, \ldots, c_n\})_k$
**3** Generated sample $\leftarrow D_y(\xi; z)$

---

Figure 21: **The generation process of RF+LZN (with class labels).** In this case, RF+LZN can simultaneously support unconditional and conditional generation. **Left:** Unconditional generation, where the LZN latent is drawn from the prior Gaussian distribution, which is exactly the same as Fig. 8. **Right:** Conditional generation, where the LZN latent is drawn from the latent zone of the corresponding class. The changes on top of unconditional generation are highlighted in gray.

distribution. To address this, we modify the $\pi_1$ distribution when computing $\text{FM}_x(\cdot)$ to be a *weighted mixture* of Dirac delta functions centered at $E_x(x_i)$, with weights corresponding to the fraction of class $x_i$ samples in the batch. During testing, we revert to a uniform prior, as the true class distribution of the batch is not available.

**Objective.** We use the version without log (Eq. (2)). We also tried the version with log (Eq. (5)) and did not observe a large difference in results.

### E.3 More Experimental Settings

**Datasets.** We use the `CIFAR10` dataset discussed in App. C.

**Metrics.** In addition to the metrics discussed in App. C, we evaluate on `CIFAR10` classification accuracy. The accuracy is evaluated on `CIFAR10` test set.

**Sampler.** Same as App. C.

**Hyperparameters.**

- `CIFAR10`
  - RF:
  - Batch size: 2000
  - Optimizer: Adam
  - Decoder learning rate: 0.002

---

**Algorithm 9:** RF+LZN classification

---

**Input** : Image encoder: $D_x$
Label decoder: $D_x$
Images: $y_1, \ldots, y_B$

**1** $z_1, \ldots, z_B \leftarrow C(y_1, \ldots, y_B)$
**2** $c_i \leftarrow D_x(z_i)$

---

Table 7: FID with different implementations for conditional image generation on CIFAR10. "CM" denotes consistency models [77]; "RF" denotes Rectified Flow [53]; "clean" denotes clean FID [64]. The best results are in  gray box .

| Algo. | FID (clean)↓ | FID (RF)↓ | FID (CM)↓ |
|---|---|---|---|
| RF | 2.85 | 2.50 | 2.47 |
| RF+LZN | 2.70 | 2.42 | 2.40 |

- Gradient clipping: 1.0
- Number of parameters in decoder: 61809539
- RF+LZN:
- Batch size: 2000
- Optimizer: Adam
- Decoder learning rate: 0.002
- Encoder learning rate: 0.00005
- Label learning rate: 0.0001
- Gradient clipping: 1.0
- Latent dimension: 200
- Number of parameters in decoder: 61911939
- Number of parameters in encoder: 49790260

**Computation cost.** Excluding the computation cost of periodic evaluation (i.e., only counting the computation cost of model training), each RF+LZN experiment takes 31 hours on 16 A100 (40 GB) GPUs.

### E.4 More Results

**Generated images.** The generated images of RF and RF+LZN are in Figs. 22 and 23.

**Ablation studies on FID implementation.** Same as App. C, we present the FID scores using three different implementations in Tab. 7. We see that, while the numbers are different, the relative ranking across all three implementations is consistent. Especially, RF+LZN achieves the best FID in all implementations.

**Ablation studies on sampling steps.** Following the experimental settings in App. C, we use the Euler sampler with varying numbers of sampling steps. As shown in Fig. 24, RF+LZN generally achieves better FID than the RF baseline across most settings.

**Ablation studies on classification techniques.** Here, we discuss several techniques for improving the classification results.

- Recall that latent computation (Eq. (1)) includes randomness from $\epsilon_i$ because each sample corresponds to a latent *zone*, not a single point. Empirically, for classification tasks, using the "center" of the latent zone yields better performance. Concretely, we set $\alpha = 0$ in Eq. (4) when computing latents. This is intuitive, as the center is likely farther from zone boundaries and better represents the sample.
- App. A.3 discusses that during training, we use a batch of samples rather than all samples to estimate latents for efficiency. However, during inference, where gradient computation is unnecessary and thus the overhead of large batch sizes is less critical, we can use a larger batch size to improve performance.

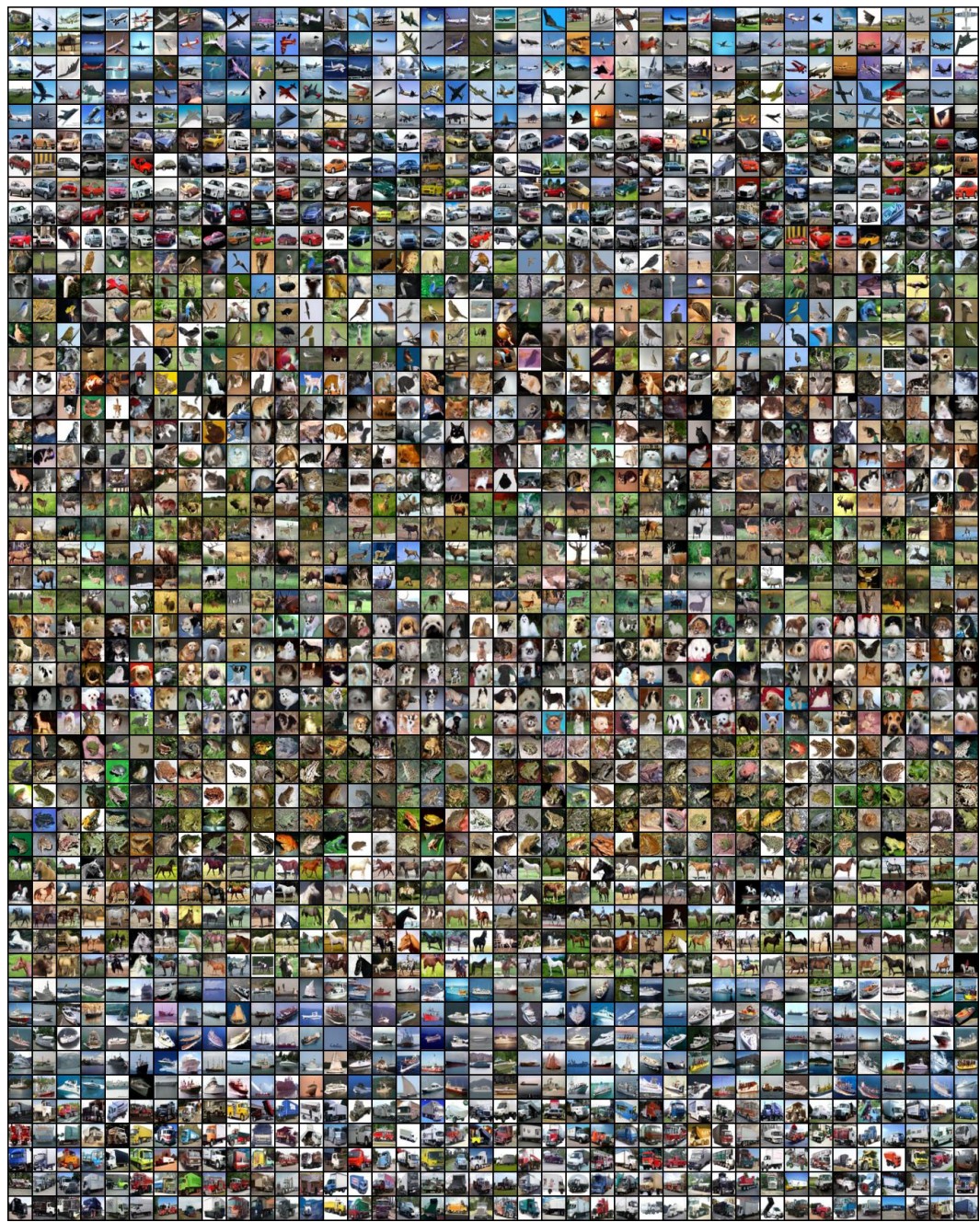

Figure 22: Generated images of RF on `CIFAR10` (conditional generation). Every 5 rows corresponds to one class in `CIFAR10`.

Fig. 25 shows that increasing the batch size and decreasing $\alpha$ improve the classification accuracy. The best setting improves the default setting (batch size$= 2000$ and $\alpha = 1.0$) by 2.9%.

**Ablation studies on latent alignment hyperparameter.** In latent alignment (§ 2.2.2), we introduced a hyperparameter $u$ that controls how many time steps are excluded from the latent alignment objective. In our main experiments, we set $u = 20$ (out of a total of 100 steps). Here, we conduct an ablation study by reducing $u$ to 5 (i.e., $4\times$ smaller). The results are shown in Tab. 8. We can see that $u$ does not affect the results much, and the performance remains better than the baseline RF across most metrics. This is expected. Unlike common hyperparameters (such as loss weights) that influence the optimal solution, $u$ does not alter the optimal solution, which is the perfect alignment

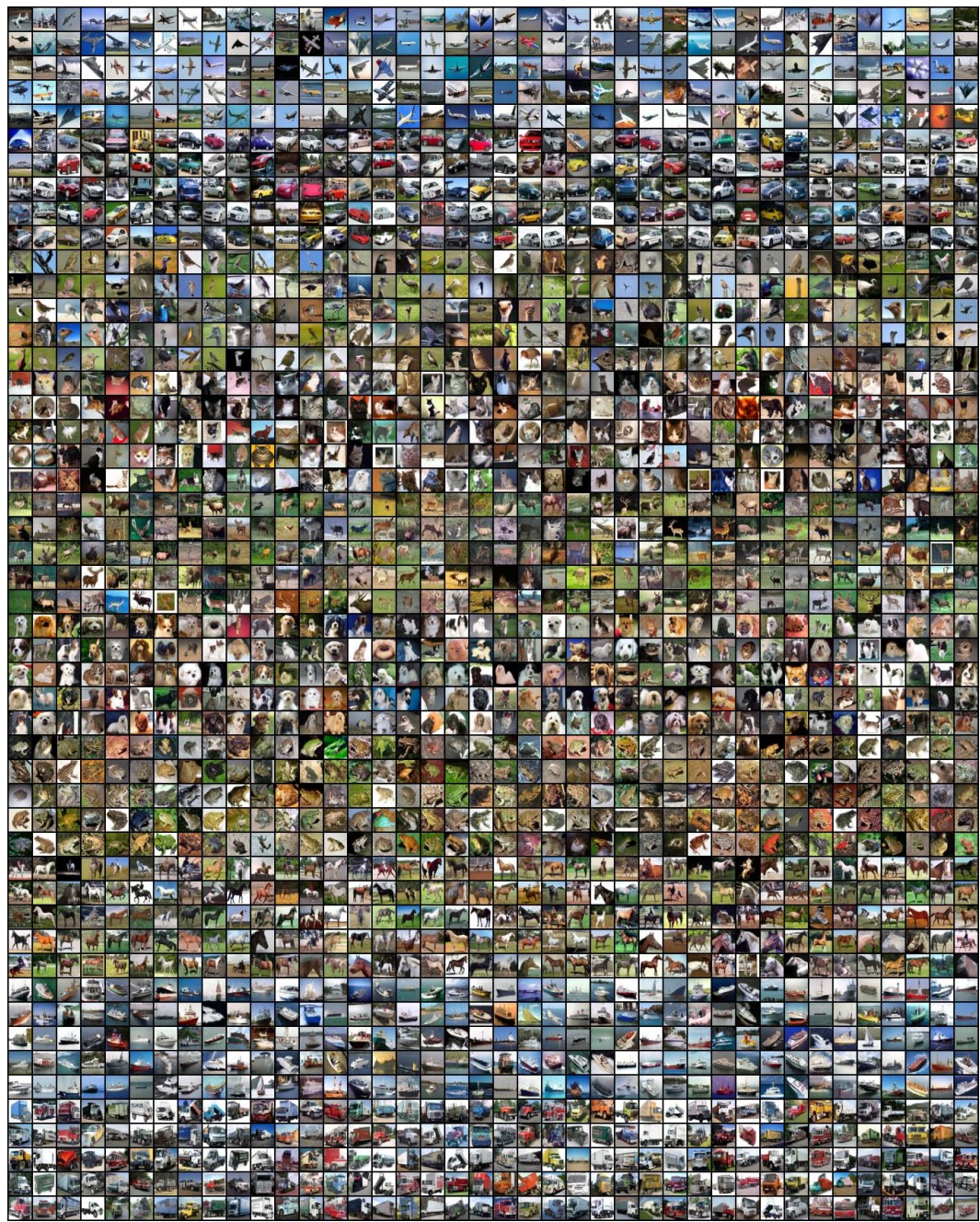

Figure 23: Generated images of RF+LZN on CIFAR10(conditional generation). Every 5 rows corresponds to one class in CIFAR10.

between two latent zones. Instead, this parameter is introduced solely to help avoid getting stuck in local optima (§ 2.2.2). We expect that any small but non-zero value of $u$ should be sufficient in practice.

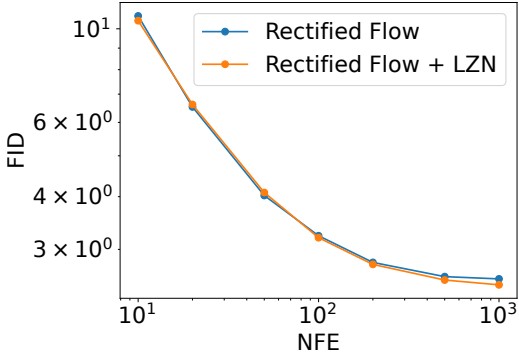

Figure 24: FID vs. number of sampling steps in the Euler sampler on CIFAR10 (conditional generation). RF+LZN outperforms RF in most cases.

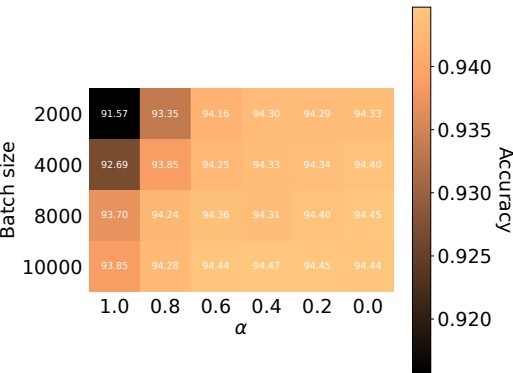

Figure 25: Classification accuracy with different hyperparameters on CIFAR10. Generally, increasing the batch size and decreasing $\alpha$ improve the classification accuracy.

Table 8: Conditional image generation quality and classification accuracy on CIFAR10. The best results are in  gray box . The hyperparameter $u$ does not impact the results much.

| Algo. | FID↓ | sFID↓ | IS↑ | Precision↑ | Recall↑ | Recon↓ | Accuracy↑ |
|---|---|---|---|---|---|---|---|
| RF | 2.47 | 4.05 | 9.77 | 0.71 | 0.58 | 0.69 | - |
| RF+LZN ($u = 20$) | 2.40 | 3.99 | 9.88 | 0.71 | 0.58 | 0.38 | 94.47 |
| RF+LZN ($u = 5$) | 2.39 | 3.99 | 9.76 | 0.71 | 0.58 | 0.36 | 94.42 |

# F  Extended Discussions on Related Work

[16] proposes to conduct flow matching in the latent space. However, it has quite different goals and techniques from LZN.

- **Goals.** The goal of [16] is to improve *generation tasks*. In contrast, our goal is more ambitious: to develop *a unified framework that supports generation, representation learning, and classification*. This broader scope requires a different design philosophy and technical approach, as detailed next.
- **Techniques.** [16] applies flow matching to the *latent space of a pre-trained Stable Diffusion autoencoder*, which is reasonable when focusing solely on generation. However, such a latent space is *high-dimensional* and retains *spatial structure*, limiting its suitability for classification and compact representation learning. To support our broader objectives, we introduce several novel techniques:
  - Match a *discrete* distribution (i.e., the anchors) to a *continuous one*, as opposed to a continuous-to-continuous distribution matching in [16].
  - Use an *adaptive latent space*, since our encoder and decoder are trained end-to-end, as opposed to using a fixed pre-trained autoencoder and fixed latent space in [16].
  - *Numerically solve* the flow directly, as opposed to training an additional model to learn the flow in [16].
  - *Latent alignment* between different data types (e.g., image and label), which is new in our paper.

# G  Extended Discussions on Limitations and Future Work

**Inference efficiency.**  It is important to note that while the training cost of LZN might be high, *at inference time, LZN is often as efficient as existing approaches.*

- For *image generation* (§ 3 and 5), we do not need to compute the latent during inference. Instead, latents are sampled from the Gaussian prior and passed directly to the decoder, making the generation speed comparable to the base model.
- For *representation learning* (§ 4), we find that dropping the final encoder layers during inference improves performance (App. D.4), similar to the observation in prior contrastive learning methods [10]. In this case, inference involves simply passing an image through the encoder *without* the latent computation process (§ 2.2.1), just like in traditional contrastive learning methods.

**Training efficiency.**  The main training bottleneck stems from the quadratic cost with respect to the batch size. Notably, this is also the case for many contrastive learning methods, including the seminal works MoCo [28] and SimCLR [10], which compute pairwise similarities between all examples in a batch.

**The parallel between LLM training and LZN training.** We observe an interesting parallel between the training of LLMs and LZN. Specifically, in LLM training, computing attention weights requires $\mathcal{O}\left(c^2 dv\right)$, where $c$ is the context length, $d$ is the attention dimension, and $v$ is the number of layers. In LZN, computing the latents (§ 2.2.1) requires $\mathcal{O}(n^2 qr)$, where $n$ is the number of samples in a batch, $q$ is the latent dimension, and $r$ is the number of solver steps. Several parallels emerge:

- Context length in LLMs ($c$) $\leftrightarrow$ Number of samples in LZN ($n$)
- Attention dimension in LLMs ($d$) $\leftrightarrow$ Latent dimension in LZN ($q$)
- Number of layers in LLMs ($v$) $\leftrightarrow$ Number of solver steps in LZN ($r$)

Not only do these parameter pairs affect the time complexity in similar ways, but their computation flows are also analogous: in LLMs, the pairwise inner product of token features is computed to derive attention weights, and these weights are computed sequentially across layers. Similarly, in LZN, the pairwise distances between intermediate anchor points of samples are computed to derive velocity, and this velocity is updated sequentially across solver steps.

While LLM training is known to be computationally expensive, recent advances have significantly improved its efficiency. Given the structural similarities, we expect that such advances in LLM training could be adapted to enhance the training efficiency of LZN as well.

**Using LZN solely to implement generative modeling.** In theory, LZN can be used solely for generative modeling. By construction (§ 3), if the decoder is trained to map latents to the corresponding data perfectly, then the generative distribution of LZN is exactly $\frac{1}{n}\sum_{i=1}^{n}\delta(s-x_i)$, i.e., the empirical distribution of the training set. We explored this approach in our early experiments. It performs well on simple datasets such as MNIST [18], but generates blurry images on more complex datasets such as CIFAR10. We hypothesize that this may be due to the minibatch approximation (App. A.3), which can break the disjoint latent property, and/or the strict requirement that latent zones have no gaps between them. We leave a deeper exploration of this direction to future work.

**Societal impacts.** Since LZN can be used to improve ML models, it has the potential for both beneficial and harmful applications. Positive use cases include creative content generation and improved information retrieval, while negative applications may involve the creation of fake or misleading content.

