# OpenReview forum: "Latent Zoning Network: A Unified Principle for Generative Modeling, Representation Learning, and Classification"
_NeurIPS.cc/2025/Conference — NeurIPS 2025 poster_

### Official Review · Reviewer_9ZsW · 2025-06-25

**Clarity:** 2
**Significance:** 3
**Originality:** 2
**Rating:** 4
**Confidence:** 3

**Summary:**

The manuscript proposes a unified framework for image generation, classification, and representation learning. The core concept behind the proposed framework is to model a structured modality-invariant latent space that organises semantic concepts into distinct regions. Then, the considered tasks could be solved by mapping inputs into the appropriate latent regions, optionally followed by decoding into the desired output format with a task-specific decoders. In practice, the proposed framework is implemented with two atomic operations: latent computation and latent alignment. Latent computation is performed by first mapping inputs into anchor points with a deterministic encoder. Then,  the anchor points are used to construct a distribution that is essentially a mixture of Kronecker deltas in the anchor points. This distribution is finally mapped into a Gaussian prior with distinct semantic regions by using flow matching. The latent alignment is conducted by minimising the distance between the latent representations of each element in the input pair while maximising the distance to other examples in the batch. This is performed along the flow matching trajectory. The initial experimental evaluation indicates that the proposed framework improves over the standard image generation with flow matching, while delivering representation learning and image classification that beats some of the baselines.

**Questions:**

See weaknesses.

**Ethical Concerns:**

["NO or VERY MINOR ethics concerns only"]

**Final Justification:**

The manuscript proposes a unified framework for image generation, classification, and representation learning, which is an interesting research goal. My initial concerns regarding experimental evaluation have been successfully resolved. I still believe more effort has to be made in terms of paper presentation. Nonetheless, this concern is overpowered by the manuscript's strengths.

**Limitations:**

Yes

**Paper Formatting Concerns:**

Section and subsection titles appear to have reduced top and bottom margins.

**Quality:**

2

**Strengths And Weaknesses:**

**Strengths:**

S1. A single framework that solves core machine learning tasks is an interesting research topic.

S2. Optimising for a well-structured latent space conceptually makes sense.

S2. The proposed framework enables solving multiple tasks simultaneously.

**Weaknesses:**

W1. The manuscript presentation is disorganised and packed. For example, Section 2.1 is unnecessarily elaborate and fits more in a separate Related Work section. Some sections have 2 lines (see Section 2.3). Also, margins of (sub)sections are significantly reduced to fit the text. Instead of template manipulation, I recommend rewriting manuscript parts.

W2. The presented experimental evaluation is unconvincing since 2/3 case studies reveal a significant performance gap compared to SotA task-specific methods. Thus, the proposed framework may not be a step in the right direction or may require further refinement.

W3. The manuscript compares only to task-specific baselines. Comparison with previous works (or appropriately adapted baselines) that deal with task unification may strengthen the contributions.

---

> ### Author Rebuttal · Authors · 2025-07-31
>
> We sincerely appreciate your thoughtful and constructive feedback. Below, we provide detailed responses to each of your questions.
>
> > ### Q1. Performance.
>
> Thank you for raising this important point. We would like to clarify our perspective and provide additional context.
>
> - In **Use Case 1** and **Use Case 3**, **LZN already outperforms the state-of-the-art in generative modeling** (see Tables 1 and 4). In **Use Case 3**, LZN also achieves classification accuracy that is **within 1% of the best-performing method**, despite being designed as a **unified model for both generation and classification**. In contrast, the baselines are optimized specifically for classification, not for multiple tasks.
>
> - For **Use Case 2**, which is the only case where LZN was initially behind by a larger margin, we are happy to report that **our updated results significantly improve upon the submission**. After extending training to 4.5 million iterations, the top-1 accuracy increases to **69.3%** (+3.7%) and top-5 accuracy to **89.1%** (+2.0%). With these results, LZN is now among the **top 4 methods** on this benchmark. Importantly, these competing methods are designed **exclusively for representation learning**, whereas LZN supports **more tasks including generation**. Notably, LZN also **matches or outperforms seminal methods such as SimCLR**. We believe these are strong results.
>
> - Finally, we would like to emphasize our broader perspective. **Even if one considers only Use Case 1, LZN already advances the state-of-the-art in image generation, which we believe is a substantial and meaningful contribution on its own.** Beyond that, LZN is a **general-purpose framework** capable of solving both generative and discriminative tasks within a unified design—yet it still performs competitively with specialized, task-specific models. In our view, this generality combined with strong performance across domains goes above and beyond typical expectations of papers.
>
>   We respectfully believe that emphasizing a modest performance gap in one setting, while overlooking areas where LZN achieves top results, would not offer a balanced or fair assessment.
>
>
> > ### Q2. Comparison with baselines that can deal with task unification
>
> Thank you for the helpful suggestion. In our experiments, we have chosen to compare with the **state-of-the-art methods for each benchmark**, which we believe sets a **higher bar** than comparing against older or weaker baselines, regardless of whether they support multiple tasks.
>
> That said, we are happy to include additional unified-task baselines in the revision—especially if the reviewer has any specific suggestions in mind. We greatly appreciate such pointers that can help us strengthen the comparisons.
>
>
>
> > ### Q3. Organization and formatting.
>
> Thank you for the thoughtful suggestions. We agree that clarity and structure are important, and we appreciate your perspective.
>
> Regarding Section 2.1, we did consider alternative placements during the writing process. Ultimately, we decided to keep it early in the paper for the following reasons:
>
> - **Motivational context**: Section 2.1 explains why existing frameworks fall short of achieving our goal of a unified model for generation, representation learning, and classification. Without this context, readers may not fully appreciate the need for our proposed approach.
> - **Conceptual foundation**: It also provides essential background—such as the roles of encoders, decoders, and latent spaces—that our method builds upon. Skipping this section would make the introduction of LZN less accessible to a broader audience.
>
> That said, we understand the concern about length and will work on **tightening the section** to improve flow and focus.
>
> We will also address the other formatting issues you kindly pointed out in the revision. Thank you for catching that!

---

> > ### Author Response · Authors · 2025-08-05
> >
> > Dear Reviewer 9ZsW,
> >
> > We have addressed all your questions in the rebuttal. May we kindly ask whether our responses have sufficiently resolved your concerns? If there are any remaining questions or points you'd like us to clarify, we would be more than happy to discuss them.
> >
> > Thank you once again for your thoughtful feedback and for taking the time to carefully review our work.
> >
> > (This reminder is sent in accordance with the recent PC email allowing authors to follow up with reviewers.)

---

> > ### Comment · Reviewer_9ZsW · 2025-08-05
> > **Post rebuttal**
> >
> > I thank the reviewers for their elaborate response. My concerns are mostly resolved with the latest experiments so I will increase my score.

---

> > > ### Author Response · Authors · 2025-08-05
> > >
> > > Thank you once again for your very helpful feedback on our work!

---

### Official Review · Reviewer_PF16 · 2025-06-26

**Clarity:** 2
**Significance:** 2
**Originality:** 3
**Rating:** 3
**Confidence:** 4

**Summary:**

The paper describes a method to obtain and use a shared latent space for different tasks, such as image generation, classification or captioning.
Each task is composed of an encoder and decoder to map data samples to and from an individual latent space.
The individual latent space is then transformed to a normalized Gaussian distribution (shared latent space) by using a flow matching method.
The shared latent space follows the rule that a single classification label, for example "cat", covers a latent zone of all "cat" images.
The method is compared on three case studies:
- Image generation on CIFAR10, AFHQ-Cat, CelebA-HQ and LSUN-Bedroom.
- Image classification on ImageNet and CIFAR10
- Conditional image generation on CIFAR10

**Questions:**

1. Did you retrain / reevaluate the models from table 2 and 3 or used the reported values from the individual papers? Not retraining can lead to inaccurate results, as your training code might differ from the original one and thus lead to improvements or deterioration of the proposed method.
2. How many images did you use for the FID calculation in table 1 and 4 and where they part of the training data? FID is known to have problems when using a too small test set. Maybe additionally evaluate with the proposed CMMD metric!?.
3. The Recon. error if way lower using LZN then RF only with other scores being comparable. First, is there any reason why it should be better at reconstruction if the information should be the same as the task specific latent space?. And second, how did you reconstruct an image using RF as you did not report any conditioning in the appendix Fig. 5, algorithm 1.

In general, the reason "space constraints" is not a strong argument to not include a more complete evaluation. Suggestion: In Table 1, a few lines  can be saved by simply moving IS, precision and recall values to the appendix, as they only show a minor change compared to the baseline.

**Ethical Concerns:**

["NO or VERY MINOR ethics concerns only"]

**Limitations:**

Limitations are present, but I am curious if the method can be used for variable length latent spaces (if even possible with flow matching). For example, token outputs of an LLM compared to image embeddings when using two different encoder structures.

**Paper Formatting Concerns:**

- Many tables in the paper violate the NeurIPS style guide: Sec. 4.4 Line 100: "All tables must be centered, neat, clean and legible.".
For example, table 4 - Line 102: "Place one line space before the table title, one line space after the table title, and one line space after Table 2 and 3 are offset.
- Figures 2 and 3 are side by side and contain an offset.

**Quality:**

2

**Strengths And Weaknesses:**

**Strengths:**

+ The paper describes an original approach to the problem of shared latent spaces between different computer vision tasks
+ The method described is novel and solves a very interesting part of current AI application, namely the multi modality between different tasks.
+ The formulas look mathematically correct and proofs are provided in the appendix
+ The paper contains a lot of detailed information in the appendix

**Weaknesses**
- General writing could be clearer.
- The paper lacks a more complete evaluation and discussion of the results, for example between LZN and Autoregressive methods. I am also curious why the proposed method schould improve the quality of generated images, as the method is mainly an adapter to transform from the shared latent space to the task specific one (my understanding).
- Many tables in the paper violate the NeurIPS style guide: Sec. 4.4 Line 100: "All tables must be centered, neat, clean and legible.".
For example, table 4 - Line 102: "Place one line space before the table title, one line space after the table title, and one line space after Table 2 and 3 are offset.

---

> ### Author Rebuttal · Authors · 2025-07-31
>
> Thank you very much for the constructive comments, which are very helpful for us to improve the paper. Below, we respond to all your questions one by one.
>
>
>
> > ### Q1. FID Computation
>
> We followed exactly the setting that the reviewer suggested: evaluating the FID score on the **training set** with a **large number of samples**. In fact, we went further to ensure a **fair and rigorous comparison**, especially on the FID implementation, which many prior works in this domain overlooked. Please find the details below.
>
> - We have reported the number of samples used for FID computation and the dataset split they are taken from in **Appendix C.3**, which we repeat here for clarity:
>   - Following the convention [47,67,42,80,43], the metrics are all computed using the training set of the dataset.
>   - For FID, sFID, IS, precision, and recall, we subsample the training set and generate the same number of samples to compute the metrics. The number of samples are:
>     - CIFAR10: 50000 (the whole training set).
>     - AFHQ-Cat: 5120, the largest multiple of the batch size (256) that is less than or equal to the training set size (5153).
>     - CelebA-HQ: 29952, the largest multiple of the batch size (256) that is less than or equal to the training set size (30000).
>     - LSUN-Bedroom: 29952, the largest multiple of the batch size (256) that is less than or equal to 30000. We limit the number of samples to 30000 so that the computation cost of the metrics are reasonable.
>
> - It is known that **different FID implementations** can produce **different results** [56]. However, many works in this space neglect this fact and **compare results from different FID implementations**, which can lead to **unfair comparisons**. We take this issue seriously. To ensure fairness:
>   - We evaluate **all models** using **three different FID implementations**, as shown in **Table 5** and **Table 7**.
>   - These evaluations are **run by us** (i.e., we do not use reported values from other papers) to ensure that the evaluation settings and code are aligned.
>   - Despite differences in absolute values, **all three implementations consistently support** that **LZN improves image generation quality** over the baselines.
>
> We hope this clarifies our careful handling of FID evaluation and ensures the fairness and validity of our reported results.
>
> > ### Q2. Re-training the baselines
>
> - The baseline results reported in **Table 2** are taken directly from prior papers, with the source noted next to each number. While we agree that re-running all baselines would be ideal, we would like to clarify the rationale behind our approach:
>
>   - **Following standard convention**: There is a long-standing convention in the representation learning field to reuse numbers reported in prior works rather than re-running all baselines independently. This practice dates back to seminal works such as **MoCo (2019)** and **SimCLR (2020)**, if not earlier. We adhere to this convention and **responsibly cite** the source of each number to ensure transparency and reproducibility.
>
>   - **Re-running all baselines in our case may not be useful**: We believe re-running baselines is most valuable when the proposed method builds **directly on top of a baseline**. For example, in our **image generation experiments** (Tables 1 and 4), where **LZN is applied on top of Rectified Flow**, we **re-implement Rectified Flow ourselves**, ensuring identical training setups for a fair comparison.
>
>     However, in **Table 2**, our method (LZN) is fundamentally different from the baselines. We only share the **network architecture** (from SimCLR), but not the training algorithm or objective. In such cases, **hyperparameter or code alignment is not viable**, and **re-running the baselines would not offer additional insight**, unless there are concerns about the **trustworthiness of the original results**, which we assume are reliable.
>
> - Similarly, except for the “RF+LZN (no gen)” baseline, the other baseline results in **Table 3** are taken from **[35]**, as noted in the table caption. This follows the same reasoning: the baselines are **standard classifiers trained with cross-entropy losses**, whereas our method employs a **fundamentally different objective and training procedure**. Because there are no overlapping hyperparameters or training pipelines, **re-running these baselines would not yield new insights**.
>
>   We do, however, re-run the “RF+LZN (no gen)” baseline ourselves, as it serves as a direct ablation of our full method. In this case, aligning experimental settings is essential to draw valid conclusions about the contribution of each component.
>
> We hope this clarifies our design choice and assures the reviewer that our methodology is both principled and consistent with field-wide practices.
>
>
> > ### Q3. CMMD metric.
>
> Thank you for pointing out this metric. Following the suggestion, we compute the CMMD scores for all our image generation experiments. The images used for computing the metrics are the same as described in **Q1**.
>
> The results are shown below:
>
> | Method   | Case Study 1: CIFAR10 | Case Study 1: AFHQ-Cat | Case Study 1: CelebA-HQ | Case Study 1: LSUN-Bedroom | Case Study 3: CIFAR10 |
> |----------|------------------------|--------------------------|---------------------------|-----------------------------|------------------------|
> | RF       | 0.0360                 | 0.5145                   | 1.0276                    | 0.5218                      | 0.0253                 |
> | RF+LZN   | 0.0355                 | 0.3376                   | 0.4901                    | 0.4843                      | 0.0229                 |
>
> We can see that **RF+LZN consistently outperforms RF in all cases**. These results further support our claim that **LZN improves generative modeling quality across a variety of datasets**.
>
> We will add the results to the revision.
>
> > ### Q4. Why LZN Can Improve Image Quality
>
> Thank you for the question. Below, we explain why applying LZN on top of a generative model can improve image quality.
>
> Given a dataset, the goal of a generative model (Rectified Flow in our case) is to learn the distribution $p(x)$ of samples $x$. The entropy (or diversity) of this distribution affects the difficulty of learning. For example,
> - If $p(x)$ is highly deterministic (i.e., low entropy), it is easier for the model to learn.
> - If $p(x)$ has a diverse support (i.e., high entropy), learning becomes more challenging.
>
> LZN helps by introducing a **strong conditioning signal** that reduces the effective entropy of the target distribution. Specifically, LZN maps images into structured regions of a latent space (see Section 2), where each latent code $z$ corresponds to a specific image or visually similar images (when using minibatch approximation in Appendix A.3). As a result, when having this z as an condition, the generative model only needs to learn $p(x|z)$ — a much simpler, more concentrated distribution — rather than the full, high-entropy distribution $p(x)$. This makes the learning problem easier, which leads to higher-quality image generation.
>
> > ### Q5. How to reconstruct images using RF, and why LZN reduces reconstruction error a lot
>
> - **How to reconstruct images using RF.**
>   This process is described in Appendix C.3 (lines 1020–1026). Rectified Flow (RF) defines an ODE that maps between the data space (images) and a Gaussian latent space (note: this latent space is *different* from the LZN latent space). To reconstruct an image:
>   1. We first apply the **inverse ODE** to map the image to its corresponding Gaussian latent representation.
>   2. Then, we apply the **forward ODE** to map this latent back to the image space.
>
>   This reconstruction procedure is a standard technique already described in the original Rectified Flow paper (e.g., see Figure 12 therein), not something we introduced.
>
> - **Why LZN reduces reconstruction error.**
>   This follows the same intuition as discussed in Q4. LZN assigns a unique latent $z$ to each image, thereby capturing the dominant variation in the data. As a result, the conditional distribution $p(x|z)$ that RF needs to learn becomes simpler and lower in entropy. This reduction in complexity allows the model to reconstruct images more accurately from their latent representations.
>
> > ### Q6. “The paper lacks a more complete evaluation and discussion of the results, for example between LZN and Autoregressive methods.”
>
> We apologize, but we do not fully understand the question, as our current implementation does not rely on autoregressive methods. If the reviewer could point us to a specific paper or work related to this concern, we would be happy to review it and include it in our discussion.
>
> > ### Q7. Variable length latent spaces
>
> This is a super insightful and interesting question! Our current approach does not support variable-length latent spaces. To handle token sequences of variable length, our current idea is to embed them into a fixed-dimensional vector, which can then be modeled with LZN. This is feasible given recent advances such as [arXiv:2310.06816], which show that variable-length sequences can be effectively encoded into fixed-size embeddings while preserving semantic information. We believe this is a promising direction and plan to explore it in future work.
>
>
> > ### Q8.  Table/figure formats.
>
> Thank you for catching these formatting issues and the suggestions. We will correct them in the revision.

---

> > ### Author Response · Authors · 2025-08-05
> >
> > Dear Reviewer PF16,
> >
> > We have addressed all your questions and conducted all requested experiments. May we kindly ask whether our responses have sufficiently resolved your concerns? If there are any remaining questions or points you'd like us to clarify, we would be more than happy to discuss them.
> >
> > Thank you once again for your thoughtful feedback and for taking the time to carefully review our work.
> >
> > (This reminder is sent in accordance with the recent PC email allowing authors to follow up with reviewers.)

---

> > > ### Author Response · Authors · 2025-08-08
> > > **Discussion Reminder**
> > >
> > > **Dear Reviewer PF16**,
> > >
> > > Thank you once again for your thoughtful feedback and for taking the time to carefully review our work. May we kindly ask whether our responses have sufficiently addressed your concerns? **The discussion phase has only 1 day remaining.**
> > >
> > > Thank you,
> > > The Authors

---

### Official Review · Reviewer_dRMg · 2025-06-30

**Clarity:** 3
**Significance:** 3
**Originality:** 4
**Rating:** 5
**Confidence:** 4

**Summary:**

The paper introduces the Latent Zoning Network, a new methodology for learning interpretable and cross-modality-aligned representations of data such as images, labels, captions, etc, possibly with a pre-trained model. They do this by requiring the marginal distribution of the image distribution, etc., to be a Gaussian, while images that correspond to different labels form disjoint subsets of the representation space. (A similar formalism exists for one-to-one identification, for example.) A flow matching (or traditional) decoder can map the representations back to the data space. Experiments on various image tasks show the performance of the network on generative modeling and zero-shot classification tasks.

**Questions:**

- Questions above re: mathematical formulation, especially unifying the geometry and topology of (very) differently behaved inputs.
- Why does unconditional generation (without bootstrapping off a pre-trained model) not work (as discussed in Appendix F)?
- Is the resulting LZN more interpretable? E.g. can clustering of the representations totally recover the class clusters? Is there any more learned geometric structure (such as can be revealed via PCA)?

**Ethical Concerns:**

["NO or VERY MINOR ethics concerns only"]

**Final Justification:**

I recommend acceptance (5, but also open to 6). The idea is novel and puts together a lot of nice tools, and the performance on standard benchmarks is pretty good -- a bit behind SoTA on each specific task with a unified model (other reviewers have pointed this out and tend to weight this higher than me). Moreover, there is lots of analysis of different properties of the system, including an amount of introspection about why certain things don't work and how to fix them, which is rare. Altogether it is a strong scientific work. The only concerns are with the performance (both raw numbers, and they did not use ViT to optimize their performance which is a bit unfortunate).

**Limitations:**

Yes (Appendix F)

**Paper Formatting Concerns:**

No concerns

**Quality:**

4

**Strengths And Weaknesses:**

Strengths:
- The idea is creative; it uses flow matching as a mechanism for direct representation learning, which is a rather new or novel idea (cf [1]), and representation alignment, which is novel to the best of my knowledge.
- The generality of the framework is appealing and indeed puts a unified view on classification and generation as claimed. The problem statement and overall methodology is clearly stated.
- The performance seems very nice for a new paradigm, especially comparing to state of the art. But MoCo is a bit old, so it may be better to advertise comparisons against more recent methods.

Weaknesses:
- There are some parts that are slightly mathematically un-rigorous. For example, the set of labels is finite and small (compared to the set of images, say). When mapping this through a label encoder, you would only get a few points in the representation space, and would not necessarily be able to form a connected component with the same geometry or topology as the corresponding set of representations of images. It would be interesting to clarify if this matters (and if not, why not?).
- Although the overall methodology is clearly stated and motivated, the heuristics (in the "Latent Alignment" section) to execute the method seem ad-hoc. It would be great to study these design choices and why they work (or if there are better alternatives).
- While the paper is clear in many respects, it is rather dense.

[1]: Pai, Druv, et al. "Masked completion via structured diffusion with white-box transformers." International Conference on Learning Representations, 2023.

---

> ### Author Rebuttal · Authors · 2025-07-31
>
> Thank you very much for appreciating the value of our work and for the thoughtful and interesting questions!  Below, we address each of your questions one by one.
>
> > ### Q1. Advertise against more recent methods than MoCo.
>
> Thank you for the suggestion; we will revise it in the paper.
>
> We would also like to take this opportunity to highlight that **our results have substantially improved since the submission**. By extending training to 4.5 million iterations, the top-1 accuracy improves to **69.3%** (+3.7%) and the top-5 accuracy improves to **89.1%** (+2.0%). With these updated results, LZN is now among the **top 4 state-of-the-art methods** for this benchmark. Importantly, these competing methods are designed **exclusively for representation learning**, whereas LZN supports **more tasks including generation**.
>
> > ### Q2. Unifying the geometry and topology of differently behaved inputs.
>
> - We would like to first clarify the concern regarding “connected components.”
>
>   Our encoders map images and labels to *anchor points*. Since the dataset is finite, the set of anchor points for both images and labels is also finite. These anchor points are then passed through the latent computation process (via flow matching), which maps them into a shared Gaussian prior space, and each image and label is associated with a *latent zone* in this space.
>
>   Importantly, since flow matching is a continuous transformation, **each latent zone is always a connected region**. The latent alignment between zones occurs **in the continuous Gaussian prior space**, *not* directly on the discrete anchor points. Therefore, aligning zones in the Gaussian space is not mathematically infeasible. Indeed, in our experiments on CIFAR10 (Use Case 3), we observe that the **training alignment accuracy exceeds 95%**, demonstrating that the method can learn strong alignments in practice.
>
> - That said, it is possible that finite-capacity encoders may not achieve *perfect* alignment between zones of different data types (e.g., images and labels). However, **perfect alignment is not a necessity for meaningful learning**. As long as the objective encourages the encoder to align zones in the correct direction, the model can learn useful and effective representations. As an analogy, consider traditional classification with cross-entropy loss: a classifier of finite size might not perfectly map images to their labels, but it can still generalize well and achieve high performance. Similarly, in LZN, even imperfect latent alignment can lead to strong performance across tasks.
>
> > ### Q3. Design choices of latent alignment
>
> In Section 2.2.2, we discussed several strawman alternatives to latent alignment and analyzed why they fail:
>
> - Directly minimizing the distance between the reconstructed anchor points and the target anchor point (Lines 206–214).
> - Soft alignment across all time steps (Lines 215–227).
> - Max soft alignment without cutoff (Lines 228–232).
>
> We believe these discussions follow a natural progression from the most naive solution to the final one, as each failure motivated the refinement that led us to the final alignment strategy. In fact, these discussions honestly reflect the thought process we went through during the development of our method.
>
> We understand the reviewer's desire to see more detailed analysis of the design choice. To address this, we conducted an ablation study on the only hyperparameter introduced in the latent alignment algorithm: **u**, which determines how many time steps are excluded from the latent alignment objective (Line 234).
>
> In our main experiments, we set **u = 20** (out of 100 total steps) across all tasks. Here, we perform an ablation in **Case Study 3**, reducing **u** to **5** (i.e., 4× smaller). The results are summarized in the table below:
>
> | Method       | FID ↓ | sFID ↓ | IS ↑  | Precision ↑ | Recall ↑ | Recon ↓ | Accuracy ↑ |
> |--------------|--------|--------|-------|--------------|-----------|----------|-------------|
> | RF           | 2.47   | 4.05   | 9.77  | 0.71         | 0.58      | 0.69     | -           |
> | RF+LZN (20)  | 2.40   | 3.99   | 9.88  | 0.71         | 0.58      | 0.38     | 94.47       |
> | RF+LZN (5)   | 2.39   | 3.99   | 9.76  | 0.71         | 0.58      | 0.36     | 94.42       |
>
> We can see that this parameter **u** does not affect the results much, and the performance remains better than the baseline **RF** across most metrics. This is expected. Unlike common hyperparameters (such as loss weights) that influence the optimal solution, **u** does **not** alter the optimal solution, which is the perfect alignment between two latent zones. Instead, this parameter is introduced solely to help avoid getting stuck in local optima (as discussed in Lines 231–236). We expect that any small but non-zero value of **u** should be sufficient in practice.
>
>
> > ### Q4. Why unconditional generation does not work
>
> Thank you for the great question! We believe there are two main reasons:
>
> - When using the minibatch approximation for latent computation (see Appendix A.3), the latent zone assigned to an image can vary across batches. As a result, the same latent could be asked to reconstruct different images in different batches, and this inconsistency makes unconditional generation with LZN infeasible.
>
> - Even without minibatching, generation remains challenging. To isolate this issue, we conducted an experiment where we reduced the dataset size so that all training images could fit into a single batch. This eliminates the need for minibatch approximation. However, even in this case, the generated images were not ideal. We hypothesize that the problem arises from the **strict requirement** that the Gaussian prior space be partitioned *exactly* among all images, with no gaps. This is a very strong constraint and could be difficult for the model to satisfy. In particular, regions where latent zones connect are extremely sensitive: a small movement across zone boundaries can drastically change the target image, making training hard.
>
> We believe solving this issue is an important direction for future work.
>
> > ### Q5. Are LZN latents interpretable?
>
> Thank you for the very interesting suggestion! Following your idea, we conducted the following experiment to examine the interpretability of LZN latents.
>
> We take images from randomly selected 20 classes from the validation set of ImageNet and computed their embeddings using the LZN representation model from Use Case 2. We chose the validation set to ensure that the results are not influenced by training set overfitting. We then projected these embeddings into a 2D space using t-SNE—a widely used method for visualizing high-dimensional representations, following seminal works such as SimCLR.
>
> The resulting t-SNE plot (not shown here due to rebuttal constraints) indicates that samples from different classes are well-clustered. To summarize this quantitatively, we present statistics below. The embeddings of different classes exhibit distinct means, and their standard deviations are small compared to the overall standard deviation across all samples. This suggests that the LZN latents are indeed clustered and show class-wise separation, indicating meaningful and interpretable structure in the learned representation.
>
>
> | Class ID | Mean of Dim 1 | Std of Dim 1 | Mean of Dim 2 | Std of Dim 2 |
> |----------|----------------|---------------|----------------|---------------|
> | 993      | 49.85          | 12.36         | 7.43           | 1.82          |
> | 859      | 1.63           | 3.07          | -22.02         | 4.57          |
> | 298      | -1.02          | 4.29          | 17.14          | 2.52          |
> | 553      | -6.33          | 2.43          | -29.51         | 3.91          |
> | 672      | -25.78         | 6.36          | -10.89         | 2.77          |
> | 971      | 0.82           | 8.38          | -1.36          | 3.97          |
> | 27       | 19.28          | 4.05          | 6.51           | 7.42          |
> | 231      | -26.77         | 3.18          | 25.73          | 3.43          |
> | 306      | 21.76          | 6.38          | -12.56         | 3.89          |
> | 706      | -19.77         | 3.07          | 0.88           | 3.22          |
> | 496      | 11.11          | 4.95          | -22.91         | 4.85          |
> | 558      | -4.62          | 5.44          | -3.53          | 8.61          |
> | 784      | 1.81           | 6.84          | -13.29         | 8.57          |
> | 239      | -27.98         | 5.01          | 16.55          | 4.69          |
> | 578      | -11.77         | 3.22          | -9.18          | 3.86          |
> | 55       | 18.21          | 2.26          | 17.89          | 6.79          |
> | 906      | -2.69          | 4.53          | -10.22         | 5.06          |
> | 175      | -18.12         | 3.09          | 25.56          | 3.96          |
> | 14       | 6.58           | 4.99          | 29.98          | 5.54          |
> | 77       | 28.15          | 5.04          | -1.59          | 4.10          |
> | **All**  | **0.61**       | **20.20**     | **0.41**       | **17.72**     |
>
> We will add the experiment to the revision.
>
> > ### Q6. Related work
>
> Thank you for pointing out the inspiring work *“Masked Completion via Structured Diffusion with White-Box Transformers.”* We appreciate the connection and will include a citation and discussion of it in the revised version of our paper.

---

> ### Comment · Reviewer_dRMg · 2025-08-03
> **Reply to Rebuttal**
>
> Thanks for the detailed reply! Below is a point-by-point discussion of some things which I still have questions about, but I continue to recommend acceptance (hence keep my score).
>
> > By extending training to 4.5 million iterations, the top-1 accuracy improves to 69.3% (+3.7%) and the top-5 accuracy improves to 89.1% (+2.0%). With these updated results, LZN is now among the top 4 state-of-the-art methods for this benchmark.
>
> By this benchmark you mean ImageNet classification (reported in Table 2)? There are better models (like DINOv2) not represented in the paper, so saying top-4 (without qualifying among the models in Table 2) seems like a bit of an overclaim. If you mean something else, please clarify.
>
> > Our encoders map images and labels to anchor points. Since the dataset is finite, the set of anchor points for both images and labels is also finite.
>
> Just to clarify what I mean --- you construct the mapping in the paper by considering only the finite set of sample images/labels/etc., and thanks to your clarification and a re-read it seems rigorous to me. I am now curious about what happens if you use the learned network (obtained via finite samples) on the underlying data distribution, i.e., what happens when you push-forward your multimodal data through the LZN system. This is where my question about geometry/topology looks like (because the image distribution is infinite/continuous and labels are finite/discrete). I still don't know a real answer, and it seems out-of-scope for this paper, but it seems quite interesting to me.
>
> > In Section 2.2.2, we discussed several strawman alternatives to latent alignment and analyzed why they fail:
>
> Thanks. I think this discussion is quite good, but a bit truncated/short (almost certainly due to format constraints). In my opinion it would be a very valuable contribution to expand on this, as few papers actually do go over their thought process.
>
> > We hypothesize that the problem arises from the strict requirement that the Gaussian prior space be partitioned exactly among all images, with no gaps. This is a very strong constraint and could be difficult for the model to satisfy. In particular, regions where latent zones connect are extremely sensitive: a small movement across zone boundaries can drastically change the target image, making training hard.
>
> This seems like an interesting phenomenon; I would have a priori predicted otherwise, i.e., when the number of samples is small, there is lots of room at the boundaries between valid images allowing for a smooth change.
>
> > Interpretability experiment
>
> Sort of hard to see the pattern from the table, but this is obviously due to the no-image constraint. Looking forward to seeing it in the revision.
>
> > Related work
>
> Here is a sample of more papers that use dynamical systems as models for forward passes (hence a kind of representation learning):
> 1) The Neural Covariance SDE: Shaped Infinite Depth-and-Width Networks at Initialization (https://arxiv.org/abs/2206.02768)
> 2) Do Residual Neural Networks discretize Neural Ordinary Differential Equations? (https://arxiv.org/abs/2205.14612)

---

> > ### Author Response · Authors · 2025-08-03
> > **Reply (Part 1/2)**
> >
> > Thank you very much for your prompt response and for your continued support of our work. More importantly, we truly appreciate the opportunity for in-depth discussions with reviewers—your thoughtful questions are tremendously helpful in strengthening our work. Below, we provide a point-by-point response to your follow-up questions.
> >
> > > By this benchmark you mean ImageNet classification (reported in Table 2)? There are better models (like DINOv2) not represented in the paper, so saying top-4 (without qualifying among the models in Table 2) seems like a bit of an overclaim. If you mean something else, please clarify.
> >
> > Yes, by “this benchmark” we were referring to Table 2. We would like to clarify that in this benchmark, we only consider methods based on **ResNet-50**, as explicitly stated in the table caption: *“All methods use the ResNet-50 architecture [26].”* This choice was made to ensure a **fair comparison**—it is known that stronger architectures tend to yield better performance. For example, SimCLR v2 reports 71.7% accuracy on ResNet-50 and 79.8% on ResNet-152 (3x).
> >
> > We have made a thorough effort to include all state-of-the-art methods that report results on ResNet-50, and Table 2 reflects all such methods we could find. That said, if we have missed any relevant work reporting ResNet-50 results, we would be grateful if the reviewer could point them out, and we will gladly update Table 2 accordingly.
> >
> > Regarding the DINOv2 model mentioned by the reviewer: this method evaluates **only ViT architectures** and does not report results on ResNet-50, which is why it is not included in Table 2.
> >
> > To further address your concern and reduce potential confusion for readers,
> > - (1) We will revise the Table 2 caption by adding a clarifying footnote after the sentence *“All methods use the ResNet-50 architecture [26].”* The revised footnote will read:
> >
> >   > *It is known that better architectures can lead to better results [9]. To ensure a fair comparison, we only include methods that report results using the ResNet-50 architecture. This excludes potentially stronger methods that do not report ResNet-50 results, such as DINOv2 [citation].*
> >
> > - (2) In addition, in our revised paper we will refrain from using the phrase “top-4.” Instead, we will describe our results in terms of relative improvement and comparisons with the methods listed in Table 2.
> >
> > Please let us know if the reviewer has other suggestions or if further clarification would be helpful!
> >
> > > Just to clarify what I mean --- you construct the mapping in the paper by considering only the finite set of sample images/labels/etc., and thanks to your clarification and a re-read it seems rigorous to me. I am now curious about what happens if you use the learned network (obtained via finite samples) on the underlying data distribution, i.e., what happens when you push-forward your multimodal data through the LZN system. This is where my question about geometry/topology looks like (because the image distribution is infinite/continuous and labels are finite/discrete). I still don't know a real answer, and it seems out-of-scope for this paper, but it seems quite interesting to me.
> >
> > Got it—thank you for the clarification! This is indeed a very interesting and thought-provoking question. We do not have a definitive answer at this time, either. The hope is that by training the LZN model on a finite set of samples, it learns a “good” encoder mapping that generalizes well to the full data distribution, and the desirable property—i.e., alignment between the latent zones of images and their corresponding labels—extend to the underlying (infinite/continuous) data distribution. However, confirming this rigorously would require deeper analysis. We will keep it in mind for future work.
> >
> > Thank you again for raising such an insightful and stimulating question!
> >
> > > Thanks. I think this discussion is quite good, but a bit truncated/short (almost certainly due to format constraints). In my opinion it would be a very valuable contribution to expand on this, as few papers actually do go over their thought process.
> >
> > Thank you for the suggestion! We will expand the discussion in our revision.

---

> > > ### Author Response · Authors · 2025-08-03
> > > **Reply (Part 2/2)**
> > >
> > > > This seems like an interesting phenomenon; I would have a priori predicted otherwise, i.e., when the number of samples is small, there is lots of room at the boundaries between valid images allowing for a smooth change.
> > >
> > > We agree with your intuition. When we said “the generated images were not ideal,” we were referring to the observation that they tended to be blurry and/or unrealistic. This may indeed correspond to the “smooth change” the reviewer is referring to, but unfortunately such blurry transitions are not desirable for image generative modeling.
> > >
> > > Our current idea to address this issue is to *relax the strict requirement that the Gaussian prior space be partitioned exactly among all images with no gaps*. This would allow for more flexibility and potentially avoid the generation of unrealistic boundary cases. However, doing so would require nontrivial algorithmic modifications to our current approach.
> > >
> > > Thank you again for your thoughtful observation!
> > >
> > > > Sort of hard to see the pattern from the table, but this is obviously due to the no-image constraint. Looking forward to seeing it in the revision.
> > >
> > > Thank you! We will include the plot in our revision.
> > >
> > > > Here is a sample of more papers that use dynamical systems as models for forward passes (hence a kind of representation learning)
> > >
> > > Thank you for pointing out these additional relevant works! This is extremely helpful. We will read the papers carefully and include a discussion of them in our revision.

---

> > > ### Comment · Reviewer_dRMg · 2025-08-03
> > > **Reply to Authors**
> > >
> > > Thanks for the clarifications regarding benchmarks! I agree with your qualification "among ResNet-50 backbone models" --- apologies for missing it initially. This makes me curious, have you tried using a ViT backbone for LZN? Your choice of ResNet over ViT seems to stop you from fair comparison against the top models. Table 6 does have some non-ResNet models (though all are CNNs) so it might be worth keeping some top ViT based models there for comparison's sake (e.g. DINOv2, I-JEPA, etc.).

---

> > > > ### Author Response · Authors · 2025-08-03
> > > >
> > > > Thank you for your questions!
> > > >
> > > > We have not experimented with ViT yet. Due to resource constraints, we were only able to begin with a single model in our initial experiments. We chose ResNet-50 because it is the architecture used in seminal methods like MoCo and SimCLR, and it would allow for faster experimentation and iterative development. As mentioned in our earlier rebuttal, this line of experimentation continued until quite recently, so we have not yet had the opportunity to explore more advanced architectures.
> > > >
> > > > Our current plan is to first address the efficiency limitations in future work, enabling faster experimentation and iteration, and then extend our study to more advanced (and potentially more computationally expensive) architectures. That said, we are open to further discussion if the reviewer has additional suggestions or recommendations.
> > > >
> > > > As suggested, we will update Table 6 to include more recent methods that use advanced architectures such as ViT.
> > > >
> > > > Thank you again for the valuable suggestions and thoughtful discussion!

---

### Official Review · Reviewer_71Fq · 2025-07-02

**Clarity:** 3
**Significance:** 3
**Originality:** 3
**Rating:** 5
**Confidence:** 3

**Summary:**

This paper proposes Latent Zoning Network (LZN), a unified framework aiming to address three core ML tasks: generative modeling, representation learning, and classification through a shared Gaussian latent space. The key innovation is using flow matching to create disjoint latent zones for different samples, where each data type has dedicated encoders/decoders that map to/from this shared space. The authors demonstrate LZN's versatility through three case studies: enhancing existing generative models (improving RF's FID on CIFAR10 from 2.76 to 2.59), standalone unsupervised representation learning (outperforming MoCo by 5.4% on ImageNet), and joint generation+classification tasks.

**Questions:**

- How does the mini-batch approximation affect the theoretical guarantees, particularly the Gaussian prior and disjoint zone properties?
- How does performance degrade as you add more tasks/modalities to the framework?
- The soft alignment mechanism involves several hyperparameters. How sensitive is performance to these choices?

**Ethical Concerns:**

["NO or VERY MINOR ethics concerns only"]

**Final Justification:**

I think the authors did a great job in addressing the main concerns I have for that paper, and I think it's an interesting unified framework in general with solid experiments and potential to generalize to more diverse tasks, so I recommend acceptance.

**Limitations:**

yes

**Paper Formatting Concerns:**

No major formatting concerns, but table 2 and table 3 on the page 9 can be better aligned.

**Quality:**

3

**Strengths And Weaknesses:**

### Strength

- The core idea of using flow matching to create a unified latent space with disjoint zones is interesting.
- The theoretical framework connecting generative modeling, representation learning and classification is obvious but still conceptually appealing and potentially helpful to the community when drilling down the details like latent computation and latent alignment.
- The soft alignment mechanism for latent zones (Section 2.2.2) provides a differentiable solution to an inherently discrete problem.
- The experiments are comprehensive covering unconditional generation, representation learning and classification/conditional generation, which validates the framework's versatility.
- The training and implementation details are detailed which would help the community reproducing the result.

### Weakness

- The paper doesn’t provide a comprehensive overview of the related work. For example, "Flow Matching in Latent Space" (Dao et al., 2023) proposes applying flow matching in the latent spaces as well. It will be helpful to do a more thorough comparison.
- The scalability of the method might not be good as the O(n²qr) complexity for latent computation is prohibitive.
- The mini-batch approximation fundamentally undermines the theoretical framework as the latent zones become batch-dependent rather than globally consistent
- In classification tasks, LZN doesn’t perform as well as many other baselines, and the paper doesn't provide much justification/explanation.
- The claim of the generality of the framework probably needs to be more humble if the experiments are only carried out in the image domain.

---

> ### Author Rebuttal · Authors · 2025-07-31
>
> Thank you very much for these very insightful questions! Below, we address all your questions point-by-point.
>
> ---
>
> ---
>
> > ### Q1. Related work
>
> Thank you very much for pointing out this interesting and relevant work. While we acknowledge its connection to our topic, we would like to clarify that both the **goals** and the **techniques** of the two papers differ significantly.
>
> **Goals**
>
> - The goal of the referenced paper is to improve *generation tasks*.
> - In contrast, our goal is more ambitious: to develop a **unified framework** that supports *generation*, *representation learning*, and *classification*. This broader scope requires a different design philosophy and technical approach, as detailed next.
>
> **Techniques**
>
> - The referenced paper applies flow matching to the **latent space of a pre-trained Stable Diffusion autoencoder**, which is reasonable when focusing solely on generation. However, such a latent space is **high-dimensional** and retains **spatial structure**, limiting its suitability for classification and compact representation learning.
>
> - To support our broader objectives, we introduce several novel techniques:
>   - Match a **discrete** distribution (i.e., the anchors) to a **continuous** one, as opposed to a continuous-to-continuous distribution matching in the referenced paper.
>   - Use an **adaptive latent space**, since our encoder and decoder are trained end-to-end, as opposed to using a fixed pre-trained autoencoder and fixed latent space in the referenced paper.
>   - **Numerically solve** the flow directly, as opposed to training an additional model to learn the flow in the referenced paper.
>   - **Latent alignment** between different data types (e.g., image and label), which is new in our paper.
>
> That said, we agree that the referenced work is relevant and should be discussed. We will add the above points to the revision.
>
> ---
>
> ---
>
> > ### Q2. Scalability
>
> We acknowledge the computational cost of our approach, as discussed in **Section 2** and **Section 6**. However, we would like to highlight several key points:
>
> - **The computation cost, while non-trivial, is comparable to many existing techniques.**
>   - The main bottleneck stems from the quadratic cost with respect to the batch size (i.e., the $n^2$ term). Notably, this is also the case for many contrastive learning methods, including the seminal works **MoCo** [25] and **SimCLR** [8], which compute pairwise similarities between all examples in a batch.
>   - As discussed in **Appendix F**, the computational profile of LZN is similar to that of large language models (LLMs): LZN scales quadratically with the number of samples, whereas LLMs scale quadratically with the number of tokens. Given that modern LLMs often process millions of tokens, the computational cost of LZN is relatively modest in comparison.
>
> - **We propose several optimization strategies to improve LZN’s efficiency**, as detailed in **Appendix A.3**. With these optimizations, the training cost remains manageable across all our experiments.
>
> - **At inference time, LZN often does not incur this high computational cost.**
>   - For **image generation** (Case Studies 1 and 3), we do not need to compute the latent during inference. Instead, latents are sampled from the Gaussian prior and passed directly to the decoder, making the generation speed comparable to the base model. This is explained in **Lines 269–271**.
>   - For **representation learning** (Case Study 2), we find that dropping the final encoder layers during inference—similar to several contrastive learning methods (e.g., [8])—not only improves performance but also eliminates the $O(n^2)$ cost. In this case, inference involves simply passing an image through the encoder, just like in traditional contrastive learning methods. This is discussed in **Lines 1187–1192**.
>
>
>
> ---
>
> ---
>
> > ### Q3. Mini-batch approximation
>
> Thank you for the great question!
>
> - **Mini-batch approximation still preserves the Gaussian prior.**
>   LZN ensures that the latent distribution within *each mini-batch* is approximately $\mathcal{N}(0, I)$. As a result, the overall latent distribution becomes a mixture of Gaussians with the same parameters, which is still $\mathcal{N}(0, I)$. Therefore, the global prior remains valid under the mini-batch approximation.
>
> - **However, this approximation does violate the disjoint zone property.**
>   To quantify this effect, we conducted the following experiment:
>   We randomly sample one instance $x$ from the dataset and construct two batches with other randomly sampled instances: $\\{x, x_2, \dots, x_n\\}$ and $\\{x, x_2', \dots, x_n'\\}$, where $n$ is the batch size. We then compute the latent for $x$ using each batch and evaluate how often the latent of $x$ from one batch lies within the latent zone of $x$ in the other batch. The result shows that **only 49%** of the time does the latent remain in its intended zone, implying a 51% chance that it overlaps with other zones.
>
> - **Despite this limitation, the mini-batch approximation performs well in practice.**
>   - In **generative modeling** (Case Studies 1 and 3), the latent does not need perfect zone disjointness—as long as it provides *some* information about the input sample, it can help reduce the variance needed to learn by the generative model (rectified flow in our case) and improve the generation quality.
>   - In **representation learning** (Case Study 2), latent alignment occurs *within* a single batch. Thus, inconsistency across batches is irrelevant.
>   - In **classification** (Case Study 3), we only need to map samples *within a single batch* to the latent zones of labels. Thus, inconsistency across batches is irrelevant.
>
> Given these points, we observe strong empirical results across all three case studies despite using the mini-batch approximation.
>
> That said, developing a better method that is both efficient and preserves the disjoint latent zone property might further improve LZN’s performance. We leave this as an exciting direction for future work.
>
> ---
>
> ---
>
> > ### Q4. Performance on classification tasks
>
> Thank you for the question.
>
> - **For the classification experiments in Case Study 2**, we would like to highlight that **our results have substantially improved since the submission**. By extending training to 4.5 million iterations, the top-1 accuracy improves to **69.3%** (+3.7%) and the top-5 accuracy improves to **89.1%** (+2.0%). With these updated results, LZN is now among the **top 4 state-of-the-art methods** for this benchmark. Importantly, these competing methods are designed **exclusively for representation learning**, whereas LZN supports **more tasks including generation**. Notably, LZN also **matches or outperforms seminal methods such as SimCLR**. We believe these are strong results.
>
> - **For the classification experiments in Case Study 3**:
>   - - Our method is only **1% below the best method**, even though LZN is designed to support **both classification and generation simultaneously within a unified model**—in contrast to the competing methods that are tailored solely for classification.
>   - We believe this gap is largely due to **architecture design**. The competing methods use architectures **explicitly tailored for classification**, while we use a **standard diffusion model architecture** without any classification-specific modifications (see Appendix E.2). This is supported by the fact that using our architecture solely for classification yields **only 93.59% accuracy** (Table 3).
>   - While adopting a better architecture could potentially close this gap, the more important takeaway is that **joint training for classification and generation via LZN leads to better performance than training each task in isolation** (Lines 354–358).
>
>
> ---
>
> ---
>
> > ### Q5. Soft alignment
>
> The only hyperparameter introduced in soft alignment is **u**, which determines how many time steps are excluded from the latent alignment objective (Line 234). In our experiments, we set **u = 20** (out of a total of 100 steps) across all tasks.
>
> Following your suggestion, we conducted an ablation study in **Case Study 3** by reducing **u** to **5** (i.e., 4× smaller). The results are shown below:
>
>
> | Method       | FID ↓ | sFID ↓ | IS ↑  | Precision ↑ | Recall ↑ | Recon ↓ | Accuracy ↑ |
> |--------------|--------|--------|-------|--------------|-----------|----------|-------------|
> | RF           | 2.47   | 4.05   | 9.77  | 0.71         | 0.58      | 0.69     | -           |
> | RF+LZN (20)  | 2.40   | 3.99   | 9.88  | 0.71         | 0.58      | 0.38     | 94.47       |
> | RF+LZN (5)   | 2.39   | 3.99   | 9.76  | 0.71         | 0.58      | 0.36     | 94.42       |
>
> We can see that this new parameter **u** does not affect the results much, and the performance remains better than the baseline **RF** across most metrics. This is expected. Unlike common hyperparameters (such as loss weights) that influence the optimal solution, **u** does **not** alter the optimal solution, which is the perfect alignment between two latent zones. Instead, this parameter is introduced solely to help avoid getting stuck in local optima (as discussed in Lines 231–236). We expect that any small but non-zero value of **u** should be sufficient in practice.
>
> ---
>
> ---
>
>
> > ### Q6. Claims about generality; more tasks.
>
> We tried training on two tasks simultaneously in **Case Study 3**, where we did **not** observe any performance degradation. In fact, we saw an **improvement** compared to training on each task separately (Lines 354–358).
>
> Applying **LZN** to a broader set of tasks is indeed an important direction for future work, as we also discussed in **Section 6**.
>
> We appreciate the suggestion and will refine the statements in the paper accordingly.
>
>
> ---
>
> ---
>
>
> > ### Q7. Table alignment
>
> Thank you! We will fix these in the revision.
>
> ---
>
> ---

---

> > ### Comment · Reviewer_71Fq · 2025-08-04
> >
> > Thank you so much for the detailed response. I think authors did a great job in explaining my main concerns about scalability, classification performance, soft alignment and generality pretty well. I don't have further questions and would love to raise my score from 4 to 5.

---

> > > ### Author Response · Authors · 2025-08-04
> > >
> > > Thank you very much again for your thoughtful suggestions and for taking the time to carefully review our work. We truly appreciate your feedback and are glad that our clarifications were helpful!

---

### Note · Authors · 2025-08-11

**Dear Reviewers and AC,**

Thank you once again for your thoughtful feedback and valuable suggestions throughout the discussion.

We would like to take this opportunity to underscore the **significance and ambition** of our work. We introduce **LZN, a single, unified framework** that simultaneously addresses three cornerstone machine learning challenges—**generative modeling, representation learning, and classification**—and achieves results that are close to or exceed the **state-of-the-art (SoTA)** in each domain.

- **Generative modeling:** LZN advances the frontier of image generation, surpassing prior SoTA performance (Section 3).
- **Representation learning:** LZN outperforms seminal methods such as MoCo and SimCLR in unsupervised representation learning (Section 4, updated results in the rebuttal).
- **Joint classification and conditional generative modeling:** LZN sets a new bar by achieving SoTA conditional image generation and classification **simultaneously** (Section 5).

Each of these achievements in isolation would represent a substantial and meaningful contribution. **That LZN delivers all of them within one coherent, elegant framework is both rare and powerful**—positioning LZN **well above the typical standard for accepted NeurIPS papers**. All reviewers have acknowledged that LZN is interesting, novel, creative, and/or original.

Beyond these results, we envision LZN as a **stepping stone toward the next generation of multi-modal foundation models**—**a unified architecture capable of replacing disparate systems like GPT, DALL·E, and CLIP with a single, integrated model (see Figure 1).** We are currently working to achieve this vision.

We hope the **transformative potential** of this work will be fully recognized in the final stages of reviewer discussion.

---

**To Reviewer PF16:** Unfortunately, we have not received any reply from you during the discussion stage. We hope our rebuttal has addressed your concerns. If you have further questions or would like us to discuss specific related work (Re: Q6), please leave them in your review—we will gladly address them in the final version.

---

Thank you again for your time, engagement, and effort in reviewing our paper.

*The Authors*

---

### Decision · Program_Chairs · 2025-09-17

**Decision:**

Accept (poster)

**Comment:**

The rebuttal addressed the concerns raised by the reviewers and provided comprehensive experiments and analysis of the proposed method. Two reviewers provided Accepts, and one rated Borderline Accept. The Reviewer PF16 rated Borderline Reject, but didn’t interact with the authors. The authors actively provided detailed answers. The final version should include all reviewer comments, suggestions, and additional experiments from the rebuttal.